# A Simple Proximal Stochastic Gradient Method for Nonsmooth Nonconvex Optimization

**Zhize Li**
IIIS, Tsinghua University
zz-li14@mails.tsinghua.edu.cn

**Jian Li**
IIIS, Tsinghua University
lijian83@mail.tsinghua.edu.cn

## Abstract

We analyze stochastic gradient algorithms for optimizing nonconvex, nonsmooth finite-sum problems. In particular, the objective function is given by the summation of a differentiable (possibly nonconvex) component, together with a possibly non-differentiable but convex component. We propose a proximal stochastic gradient algorithm based on variance reduction, called ProxSVRG+. Our main contribution lies in the analysis of ProxSVRG+. It recovers several existing convergence results and improves/generalizes them (in terms of the number of stochastic gradient oracle calls and proximal oracle calls). In particular, ProxSVRG+ generalizes the best results given by the SCSG algorithm, recently proposed by [Lei et al., 2017] for the smooth nonconvex case. ProxSVRG+ is also more straightforward than SCSG and yields simpler analysis. Moreover, ProxSVRG+ outperforms the deterministic proximal gradient descent (ProxGD) for a wide range of minibatch sizes, which partially solves an open problem proposed in [Reddi et al., 2016]. Also, ProxSVRG+ uses much less proximal oracle calls than ProxSVRG [Reddi et al., 2016]. Moreover, for nonconvex functions satisfied Polyak-Łojasiewicz condition, we prove that ProxSVRG+ achieves a global linear convergence rate without restart unlike ProxSVRG. Thus, it can *automatically* switch to the faster linear convergence in some regions as long as the objective function satisfies the PL condition locally in these regions. Finally, we conduct several experiments and the experimental results are consistent with the theoretical results.

## 1 Introduction

In this paper, we consider nonsmooth nonconvex finite-sum optimization problems of the form

$$\min_x \Phi(x) := f(x) + h(x), \tag{1}$$

where $f(x) := \frac{1}{n}\sum_{i=1}^{n} f_i(x)$ and each $f_i(x)$ is possibly nonconvex with a Lipschitz continuous gradient, while $h(x)$ is nonsmooth but convex (e.g., $l_1$ norm $\|x\|_1$ or indicator function $I_C(x)$ for some convex set $C$). We assume that the proximal operator of $h(x)$ can be computed efficiently.

This above optimization problem is fundamental to many machine learning problems, ranging from convex optimization such as Lasso, SVM to highly nonconvex problem such as optimizing deep neural networks. There has been extensive research when $f(x)$ is convex (see e.g., [25, 7, 15, 1]). In particular, if $f_i$s are strongly-convex, Xiao and Zhang [25] proposed the Prox-SVRG algorithm, which achieves a linear convergence rate, based on the well-known variance reduction technique SVRG developed in [12]. In recent years, due to the increasing popularity of deep learning, the nonconvex case has attracted significant attention. See e.g., [9, 3, 23, 17] for results on the smooth nonconvex case (i.e., $h(x) \equiv 0$). Very recently, Zhou et al. [27] proposed an algorithm with stochastic gradient complexity $\widetilde{O}(\frac{1}{\epsilon^{3/2}} \wedge \frac{n^{1/2}}{\epsilon})$, improving the previous results $O(\frac{1}{\epsilon^{5/3}})$ [17] and $O(\frac{n^{2/3}}{\epsilon})$ [3]. For the more general *nonsmooth nonconvex* case, the research is still somewhat limited.

Recently, for the nonsmooth nonconvex case, Reddi et al. [24] provided two algorithms called ProxSVRG and ProxSAGA, which are based on the well-known variance reduction techniques SVRG and SAGA [12, 7]. Also, we would like to mention that Aravkin and Davis [5] considered the case when $h$ can be nonconvex in a more general context of robust optimization. Before that, Ghadimi et al. [10] analyzed the deterministic proximal gradient method (i.e., computing the full-gradient in every iteration) for nonconvex nonsmooth problems. Here we denote it as ProxGD. Ghadimi et al. [10] also considered the stochastic case (here we denote it as ProxSGD). However, ProxSGD requires the batch sizes being a large number (i.e., $\Omega(1/\epsilon)$) or increasing with the iteration number $t$. Note that ProxSGD may reduce to deterministic ProxGD after some iterations due to the increasing batch sizes. Note that from the perspectives of both computational efficiency and statistical generalization, always computing full-gradient (GD or ProxGD) may not be desirable for large-scale machine learning problems. A reasonable minibatch size is also desirable in practice, since the computation of minibatch stochastic gradients can be implemented in parallel. In fact, practitioners typically use moderate minibatch sizes, often ranging from something like 16 or 32 to a few hundreds (sometimes to a few thousands, see e.g., [11]).[1] Hence, it is important to study the convergence in moderate and constant minibatch size regime.

Reddi et al. [24] provided the first non-asymptotic convergence rates for ProxSVRG with minibatch size at most $O(n^{2/3})$, for the nonsmooth nonconvex problems. However, their convergence bounds (using constant or moderate size minibatches) are worse than the deterministic ProxGD in terms of the number of proximal oracle calls. Note that their algorithms (i.e., ProxSVRG/SAGA) outperform the ProxGD only if they use quite large minibatch size $b = O(n^{2/3})$. Note that in a typical application, the number of training data is $n = 10^6 \sim 10^9$, and $n^{2/3} = 10^4 \sim 10^6$. Hence, $O(n^{2/3})$ is a quite large minibatch size. Finally, they presented an important open problem of *developing stochastic methods with provably better performance than ProxGD with constant minibatch size*.

**Our Contribution:** In this paper, we propose a very straightforward algorithm called ProxSVRG+ to solve the nonsmooth nonconvex problem (1). Our main technical contribution lies in the new convergence analysis of ProxSVRG+, which has notable difference from that of ProxSVRG [24]. We list our results in Table 1–3, and Figure 1–2. Our convergence results are stated in terms of the number of stochastic first-order oracle (*SFO*) calls and proximal oracle (*PO*) calls (see Definition 2). We would like to highlight the following results yielded by our new analysis:

1) ProxSVRG+ is $\sqrt{b}$ (resp. $\sqrt{b}\epsilon n$) times faster than ProxGD in terms of #SFO when $b \leq n^{2/3}$ (resp. $b \leq 1/\epsilon^{2/3}$), and $n/b$ times faster than ProxGD when $b > n^{2/3}$ (resp. $b > 1/\epsilon^{2/3}$). Note that #PO $= O(1/\epsilon)$ for both ProxSVRG+ and ProxGD. Obviously, for any super constant $b$, ProxSVRG+ is strictly better than ProxGD. Hence, we partially answer the open question (i.e. developing stochastic methods with provably better performance than ProxGD with constant minibatch size $b$) proposed in [24]. Also, ProxSVRG+ matches the best result achieved by ProxSVRG at $b = n^{2/3}$, and ProxSVRG+ is strictly better for smaller $b$ (using less PO calls). See Figure 1 for an overview.

2) Assuming that the variance of the stochastic gradient is bounded, i.e. online/stochastic setting, ProvSVRG+ generalizes the best result achieved by SCSG, recently proposed by Lei et al. [17] for the smooth nonconvex case, i.e., $h(x) \equiv 0$ in form (1) (see Table 1, the 5th row). ProxSVRG+ is more straightforward than SCSG and yields simpler proof. Our results also match the results of Natasha1.5 proposed by Allen-Zhu [2] very recently, in terms of #SFO, if there is no additional assumption (see Footnote 2 for details). In terms of #PO, our algorithm outperforms Natasha1.5. We also note that SCSG [17] and ProxSVRG [24] achieved their best convergence results with $b = 1$ and $b = n^{2/3}$ respectively, while ProxSVRG+ achieves the best result with $b = 1/\epsilon^{2/3}$ (see Figure 1), which is a moderate minibatch size (which is not too small for parallelism and not too large for better generalization). In our experiments, the best $b$ for ProxSVRG and ProxSVRG+ in the MNIST experiments is 4096 and 256, respectively (see the second row of Figure 4).

3) For the nonconvex functions satisfying Polyak-Łojasiewicz condition [22], we prove that Prox-SVRG+ achieves a global linear convergence rate *without restart*, while Reddi et al. [24] used PL-SVRG to restart ProxSVRG $O(\log(1/\epsilon))$ times to obtain the linear convergence rate. Moreover, ProxSVRG+ also improves ProxGD and ProxSVRG/SAGA, and generalizes the results of SCSG in this case (see Table 3). Also see the remarks after Theorem 2 for more details.

Table 1: Comparison of the SFO and PO complexity

| Algorithms | Stochastic first-order oracle (SFO) | Proximal oracle (PO) | Additional condition |
|---|---|---|---|
| ProxGD [10] (full gradient) | $O(n/\epsilon)$ | $O(1/\epsilon)$ | – |
| ProxSGD [10] | $O(b/\epsilon)$ | $O(1/\epsilon)$ | $\sigma = O(1)$, $b \geq 1/\epsilon$ |
| ProxSVRG/SAGA [24] | $O\big(\frac{n}{\epsilon\sqrt{b}} + n\big)$ | $O\big(\frac{n}{\epsilon b^{3/2}}\big)$ | $b \leq n^{2/3}$ |
| SCSG [17] (smooth nonconvex, i.e., $h(x) \equiv 0$ in (1)) | $O\big(\frac{b^{1/3}}{\epsilon}\big(n \wedge \frac{1}{\epsilon}\big)^{2/3}\big)$ | NA | $\sigma = O(1)$ |
| Natasha1.5 [2] | $O(1/\epsilon^{5/3})$ [2] | $O(1/\epsilon^{5/3})$ | $\sigma = O(1)$ |
| ProxSVRG+ (this paper) | $O\big(\frac{n}{\epsilon\sqrt{b}} + \frac{b}{\epsilon}\big)$ | $O(1/\epsilon)$ | – |
| | $O\big(\big(n \wedge \frac{1}{\epsilon}\big)\frac{1}{\epsilon\sqrt{b}} + \frac{b}{\epsilon}\big)$ | $O(1/\epsilon)$ | $\sigma = O(1)$ |

The $\wedge$ denotes the minimum and $b$ denotes the minibatch size. The definitions of SFO and PO are given in Definition 2, $\sigma$ (in the last column) is defined in Assumption 1.

Table 2: Some recommended minibatch sizes $b$

| Algorithm | Minibatch | SFO | PO | Cond. | Notes |
|---|---|---|---|---|---|
| ProxSVRG+ | $b = 1$ | $O(n/\epsilon)$ | $O(1/\epsilon)$ | – | Same as ProxGD |
| | | $O(1/\epsilon^2)$ | $O(1/\epsilon)$ | $\sigma = O(1)$ | Same as ProxSGD |
| | $b = \frac{1}{\epsilon^{2/3}}$ | $O\big(\frac{n}{\epsilon^{2/3}} + \frac{1}{\epsilon^{5/3}}\big)$ | $O(1/\epsilon)$ | – | Better than ProxGD, does not need $\sigma = O(1)$ |
| | | $O\big(\frac{1}{\epsilon^{5/3}}\big)$ | $O(1/\epsilon)$ | $\sigma = O(1)$, $n > 1/\epsilon$ | Better than ProxGD and ProxSVRG/SAGA, same as SCSG (in SFO) |
| | $b = n^{2/3}$ | $O\big(\frac{n^{2/3}}{\epsilon}\big)$ | $O(1/\epsilon)$ | – | Same as ProxSVRG/SAGA |
| | $b = n$ | $O(n/\epsilon)$ | $O(1/\epsilon)$ | – | Same as ProxGD |

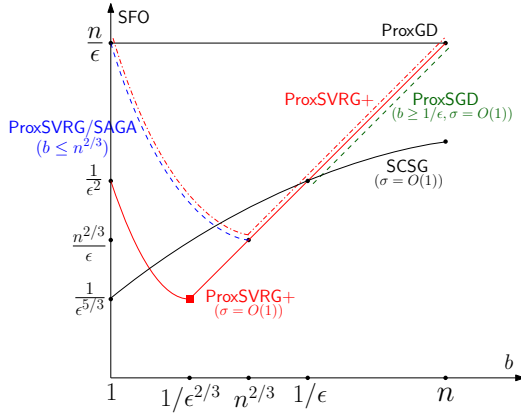

Figure 1: SFO complexity in terms of minibatch $b$ [3]

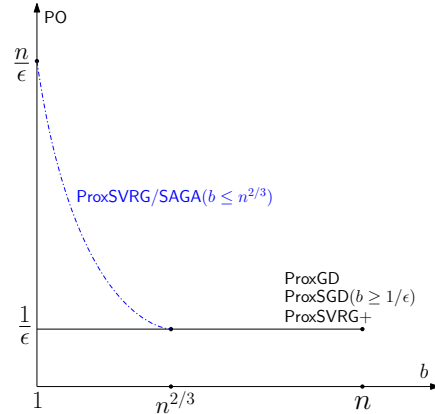

Figure 2: PO complexity in terms of minibatch $b$

[2] Natasha 1.5 used an additional parameter, called strongly nonconvex parameter $\widetilde{L}$ ($\widetilde{L} \leq L$) and #SFO in [2] is $O(\frac{1}{\epsilon^{3/2}} + \frac{\widetilde{L}^{1/3}}{\epsilon^{5/3}})$. If $\widetilde{L}$ is much smaller than $L$, the bound is better. Without any additional assumption, the default value of $\widetilde{L}$ is $L$. The result listed in the table is the $\widetilde{L} = L$ case. Besides, one can verify that #PO of Natasha1.5 is the same as its #SFO.

[3] Note that the curve of ProxSGD overlaps with ProxSVRG+ for $b \geq 1/\epsilon$, and the curve of ProxSVRG/SAGA overlaps with ProxSVRG+ for $b \leq n^{2/3}$ in Figure 1. We did not plot Natasha 1.5 since it did not consider the minibatch case, i.e., $b \equiv 1$ in Natasha 1.5.

## 2  Preliminaries

We assume that $f_i(x)$ in (1) has an $L$-Lipschitz continuous gradient for all $i \in [n]$, i.e., there is a constant $L$ such that

$$\|\nabla f_i(x) - \nabla f_i(y)\| \leq L\|x - y\|, \tag{2}$$

where $\|\cdot\|$ denotes the Eculidean norm $\|\cdot\|_2$. Note that $f_i(x)$ does not need to be convex. We also assume that the nonsmooth convex function $h(x)$ in (1) is well structured, i.e., the following proximal operator on $h$ can be computed efficiently:

$$\text{prox}_{\eta h}(x) := \arg\min_{y \in \mathbb{R}^d} \left( h(y) + \frac{1}{2\eta}\|y - x\|^2 \right). \tag{3}$$

For convex problems, one typically uses the optimality gap $\Phi(x) - \Phi(x^*)$ as the convergence criterion (see e.g., [21]). But for general nonconvex problems, one typically uses the gradient norm as the convergence criterion. E.g., for smooth nonconvex problems (i.e., $h(x) \equiv 0$), Ghadimi and Lan [9], Reddi et al. [23] and Lei et al. [17] used $\|\nabla \Phi(x)\|^2$ (i.e., $\|\nabla f(x)\|^2$) to measure the convergence results. In order to analyze the convergence results for *nonsmooth* nonconvex problems, we need to define the *gradient mapping* as follows (as in [10, 24]):

$$\mathcal{G}_\eta(x) := \frac{1}{\eta}\Big( x - \text{prox}_{\eta h}\big( x - \eta \nabla f(x) \big) \Big). \tag{4}$$

We often use an equivalent but useful form of $\text{prox}_{\eta h}\big( x - \eta \nabla f(x) \big)$ as follows:

$$\text{prox}_{\eta h}\big( x - \eta \nabla f(x) \big) = \arg\min_{y \in \mathbb{R}^d} \left( h(y) + \frac{1}{2\eta}\|y - x\|^2 + \langle \nabla f(x), y \rangle \right). \tag{5}$$

Note that if $h(x)$ is a constant function (in particular, zero), this gradient mapping reduces to the ordinary gradient: $\mathcal{G}_\eta(x) = \nabla \Phi(x) = \nabla f(x)$. In this paper, we use the gradient mapping $\mathcal{G}_\eta(x)$ as the convergence criterion (same as [10, 24]).

**Definition 1** $\hat{x}$ *is called an $\epsilon$-accurate solution for problem (1) if* $\mathbb{E}[\|\mathcal{G}_\eta(\hat{x})\|^2] \leq \epsilon$, *where $\hat{x}$ denotes the point returned by a stochastic algorithm.*

Note that the metric $\mathcal{G}_\eta(x)$ has already normalized the step-size $\eta$, i.e., it is independent of different algorithms. Also it is indeed a convergence metric for $\Phi(x) = f(x) + h(x)$. Let $x^+ := \text{prox}_{\eta h}\big( x - \eta \nabla f(x) \big)$, then $\mathcal{G}_\eta(x) := \frac{1}{\eta}\big( x - x^+ \big)$. If $\|\mathcal{G}_\eta(x)\| = \frac{1}{\eta}\|x - x^+\| = \|\nabla f(x) + \partial h(x^+)\| \leq \epsilon$, then $\|\partial \Phi(x^+)\| = \|\nabla f(x^+) + \partial h(x^+)\| \leq L\|x - x^+\| + \|\nabla f(x) + \partial h(x^+)\| \leq L\eta\epsilon + \epsilon = O(\epsilon)$. Thus the next iteration point $x^+$ is an $\epsilon$-approximate stationary solution for the objection function $\Phi(x) = f(x) + h(x)$.

To measure the efficiency of a stochastic algorithm, we use the following oracle complexity.

**Definition 2** *(1) Stochastic first-order oracle (SFO): given a point $x$, SFO outputs a stochastic gradient $\nabla f_i(x)$ such that $\mathbb{E}_{i \sim [n]}[\nabla f_i(x)] = \nabla f(x)$.*

*(2) Proximal oracle (PO): given a point $x$, PO outputs the result of the proximal projection $\text{prox}_{\eta h}(x)$ (see (3)).*

Sometimes, the following assumption on the variance of the stochastic gradients is needed (see the last column "additional condition" in Table 1). Such an assumption is necessary if one wants the convergence result to be independent of $n$. People also denote this case as the online/stochastic setting, in which the full gradient is not available (see e.g., [2, 16]).

**Assumption 1** *For $\forall x$, $\mathbb{E}[\|\nabla f_i(x) - \nabla f(x)\|^2] \leq \sigma^2$, where $\sigma > 0$ is a constant and $\nabla f_i(x)$ is a stochastic gradient.*

## 3  Nonconvex ProxSVRG+ Algorithm

In this section, we propose a proximal stochastic gradient algorithm called ProxSVRG+, which is very straightforward (similar to nonconvex ProxSVRG [24] and convex Prox-SVRG [25]). The details are described in Algorithm 1. We call $B$ the batch size and $b$ the minibatch size.

**Algorithm 1** Nonconvex ProxSVRG+

---

**Input:** initial point $x_0$, batch size $B$, minibatch size $b$, epoch length $m$, step size $\eta$
1: $\widetilde{x}^0 = x_0$
2: **for** $s = 1, 2, \ldots, S$ **do**
3:     $x_0^s = \widetilde{x}^{s-1}$
4:     $g^s = \frac{1}{B} \sum_{j \in I_B} \nabla f_j(\widetilde{x}^{s-1})$ [4]
5:     **for** $t = 1, 2, \ldots, m$ **do**
6:        $v_{t-1}^s = \frac{1}{b} \sum_{i \in I_b} \left( \nabla f_i(x_{t-1}^s) - \nabla f_i(\widetilde{x}^{s-1}) \right) + g^s$
7:        $x_t^s = \text{prox}_{\eta h}(x_{t-1}^s - \eta v_{t-1}^s)$ (call PO once)
8:     **end for**
9:     $\widetilde{x}^s = x_m^s$
10: **end for**
**Output:** $\hat{x}$ chosen uniformly from $\{x_{t-1}^s\}_{t \in [m], s \in [S]}$

---

Compared with Prox-SVRG, ProxSVRG [24] analyzed the nonconvex functions while Prox-SVRG [25] only analyzed the convex functions. The major difference of our ProxSVRG+ is that we avoid the computation of the full gradient at the beginning of each epoch, i.e., $B$ may not equal to $n$ (see Line 4 of Algorithm 1) while ProxSVRG and Prox-SVRG used $B = n$. Note that even if we choose $B = n$, our analysis is more stronger than ProxSVRG [24]. Also, our ProxSVRG+ shows that the "stochastically controlled" trick of SCSG [17] (i.e., the length of each epoch is a geometrically distributed random variable) is not really necessary for achieving the desired bound.[5] As a result, our straightforward ProxSVRG+ generalizes the result of SCSG to the more general nonsmooth nonconvex case and yields simpler analysis.

## 4  Convergence Results

Now, we present the main theorem for our ProxSVRG+ which corresponds to the last two rows in Table 1 and give some remarks.

**Theorem 1** *Let step size $\eta = \frac{1}{6L}$ and $m = \sqrt{b}$, where $b$ denotes the minibatch size. Then $\hat{x}$ returned by Algorithm 1 is an $\epsilon$-accurate solution for problem (1) (i.e., $\mathbb{E}[\|\mathcal{G}_\eta(\hat{x})\|^2] \le \epsilon$). We distinguish the following two cases:*

*1) We let batch size $B = n$. The number of SFO calls is at most*

$$36L\big(\Phi(x_0) - \Phi(x^*)\big)\Big(\frac{B}{\epsilon\sqrt{b}} + \frac{b}{\epsilon}\Big) = O\Big(\frac{n}{\epsilon\sqrt{b}} + \frac{b}{\epsilon}\Big).$$

*2) Under Assumption 1, we let batch size $B = \min\{6\sigma^2/\epsilon, n\}$. The number of SFO calls is at most*

$$36L\big(\Phi(x_0) - \Phi(x^*)\big)\Big(\frac{B}{\epsilon\sqrt{b}} + \frac{b}{\epsilon}\Big) = O\Big(\big(n \wedge \frac{1}{\epsilon}\big)\frac{1}{\epsilon\sqrt{b}} + \frac{b}{\epsilon}\Big),$$

*where $\wedge$ denotes the minimum.*

*In both cases, the number of PO calls equals to the total number of iterations $T$, which is at most*

$$\frac{36L}{\epsilon}\big(\Phi(x_0) - \Phi(x^*)\big) = O\left(\frac{1}{\epsilon}\right).$$

**Remark:** The proof for Theorem 1 is notably different from that of ProxSVRG [24]. Reddi et al. [24] used a Lyapunov function $R_t^{s+1} = \Phi(x_t^{s+1}) + c_t\|x_t^{s+1} - \widetilde{x}^S\|^2$ and showed that $R^s$ decreases by

the accumulated gradient mapping $\sum_{t=1}^m \|\mathcal{G}_\eta(x_t^s)\|^2$ in epoch $s$. In our proof, we directly show that $\Phi(x^s)$ decreases by $\sum_{t=1}^m \|\mathcal{G}_\eta(x_t^s)\|^2$ using a different analysis. This is made possible by tightening the inequalities using Young's inequality and Lemma 2 (which gives the relation between the variance of stochastic gradient estimator and the inner product of the gradient difference and point difference). Also, our convergence result holds for any minibatch size $b \in [1, n]$ unlike ProxSVRG $b \le n^{2/3}$ (see Fig. 1). Moreover, ProxSVRG+ uses much less proximal oracle calls than ProxSVRG (see Fig. 2).

For the online/stochastic Case 2), we avoid the computation of the full gradient at the beginning of each epoch, i.e., $B \ne n$. Then, we use the similar idea in SCSG [17] to bound the variance term, but we do not need the "stochastically controlled" trick of SCSG (as we discussed in Section 3) to achieve the desired convergence bound which yields a much simpler analysis for our ProxSVRG+.

We defer the proof of Theorem 1 to Appendix A.1. Also, similar convergence results for other choices of epoch length $m \ne \sqrt{b}$ are provided in Appendix A.2.

## 5 Convergence Under PL Condition

In this section, we provide the global linear convergence rate for nonconvex functions under the Polyak-Łojasiewicz (PL) condition [22]. The original form of PL condition is

$$\exists \mu > 0, \text{ such that } \|\nabla f(x)\|^2 \ge 2\mu(f(x) - f^*), \ \forall x, \tag{6}$$

where $f^*$ denotes the (global) optimal function value. It is worth noting that $f$ satisfies PL condition when $f$ is $\mu$-strongly convex. Moreover, Karimi et al. [13] showed that PL condition is weaker than many conditions (e.g., strong convexity (SC), restricted strong convexity (RSC) and weak strong convexity (WSC) [20]). Also, if $f$ is convex, PL condition is equivalent to the error bounds (EB) and quadratic growth (QG) condition [19, 4]. Note that PL condition implies that every stationary point is a global minimum, but it does not imply there is a unique minimum unlike the strongly convex condition.

Due to the nonsmooth term $h(x)$ in problem (1), we use the gradient mapping (see (4)) to define a more general form of PL condition as follows:

$$\exists \mu > 0, \text{ such that } \|\mathcal{G}_\eta(x)\|^2 \ge 2\mu(\Phi(x) - \Phi^*), \ \forall x. \tag{7}$$

Recall that if $h(x)$ is a constant function, the gradient mapping reduces to $\mathcal{G}_\eta(x) = \nabla\Phi(x) = \nabla f(x)$. Our PL condition is different from the one used in [13, 24]. See the Remark (3) after Theorem 2.

**Further Motivation:** In many cases, although the loss function is generally nonconvex, the local region near a local minimum may satisfy the PL condition. In fact, there are some recent studies showing the strong convexity in the neighborhood of the ground truth solution in some simple neural networks [26, 8]. Such results provide further motivation for studying the PL condition. Moreover, we argue that our ProxSVRG+ is particularly desirable in this case since it first converges sublinearly $O(1/\epsilon)$ (according to Theorem 1) then *automatically* converges linearly $O(\log 1/\epsilon)$ (according to Theorem 2) in the regions as long as the loss function satisfies the PL condition locally in these regions. We list the convergence results in Table 3 (also see the remarks after Theorem 2).

Table 3: Under the PL condition with parameter $\mu$

| Algorithms | Stochastic first-order oracle (SFO) | Proximal oracle (PO) | Addi. condition |
|---|---|---|---|
| ProxGD [13] (full gradient) | $O(\frac{n}{\mu}\log\frac{1}{\epsilon})$ | $O(\frac{1}{\mu}\log\frac{1}{\epsilon})$ | – |
| ProxSVRG/SAGA [24] | $O\big(\frac{n}{\mu\sqrt{b}}\log\frac{1}{\epsilon} + n\log\frac{1}{\epsilon}\big)$ | $O\big(\frac{n}{\mu b^{3/2}}\log\frac{1}{\epsilon}\big)$ | $b \le n^{2/3}$ |
| SCSG [17] (smooth nonconvex, i.e., $h(x) \equiv 0$) | $O\Big(\frac{b^{\frac{1}{3}}}{\mu}\big(n \wedge \frac{1}{\mu\epsilon}\big)^{\frac{2}{3}}\log\frac{1}{\epsilon} + \big(n \wedge \frac{1}{\mu\epsilon}\big)\log\frac{1}{\epsilon}\Big)$ | NA | $\sigma = O(1)$ |
| ProxSVRG+ (this paper) | $O\big(\frac{n}{\mu\sqrt{b}}\log\frac{1}{\epsilon} + \frac{b}{\mu}\log\frac{1}{\epsilon}\big)$ | $O(\frac{1}{\mu}\log\frac{1}{\epsilon})$ | – |
| | $O\Big(\big(n \wedge \frac{1}{\mu\epsilon}\big)\frac{1}{\mu\sqrt{b}}\log\frac{1}{\epsilon} + \frac{b}{\mu}\log\frac{1}{\epsilon}\Big)$ | $O(\frac{1}{\mu}\log\frac{1}{\epsilon})$ | $\sigma = O(1)$ |

The notation $\wedge$ denotes the minimum. Similar to Table 2, ProxSVRG+ is better than ProxGD and ProxSVRG/SAGA, and generalizes the SCSG by choosing different minibatch size $b$.

Similar to Theorem 1, we provide the convergence result of ProxSVRG+ (Algorithm 1) under PL-condition in the following Theorem 2. Note that under PL condition (i.e. (7) holds), ProxSVRG+ can directly use the final iteration $\widetilde{x}^S$ as the output point instead of the randomly chosen one $\hat{x}$. Similar to [24], we assume the condition number $L/\mu > \sqrt{n}$ for simplicity. Otherwise, one can choose different step size $\eta$ which is similar to the case where we deal with other choices of epoch length $m$ (see Appendix A.2).

**Theorem 2** *Let step size $\eta = \frac{1}{6L}$ and $b$ denote the minibatch size. Then the final iteration point $\widetilde{x}^S$ in Algorithm 1 satisfies $\mathbb{E}[\Phi(\widetilde{x}^S) - \Phi^*] \leq \epsilon$ under PL condition. We distinguish the following two cases:*

*1) We let batch size $B = n$. The number of SFO calls is bounded by*

$$O\Big(\frac{n}{\mu\sqrt{b}}\log\frac{1}{\epsilon} + \frac{b}{\mu}\log\frac{1}{\epsilon}\Big).$$

*2) Under Assumption 1, we let batch size $B = \min\{\frac{6\sigma^2}{\mu\epsilon}, n\}$. The number of SFO calls is bounded by*

$$O\Big(\big(n \wedge \frac{1}{\mu\epsilon}\big)\frac{1}{\mu\sqrt{b}}\log\frac{1}{\epsilon} + \frac{b}{\mu}\log\frac{1}{\epsilon}\Big),$$

*where $\wedge$ denotes the minimum.*

*In both cases, the number of PO calls equals to the total number of iterations $T$ which is bounded by*

$$O\Big(\frac{1}{\mu}\log\frac{1}{\epsilon}\Big).$$

**Remark:**

(1) We show that ProxSVRG+ directly obtains a global linear convergence rate without restart by a nontrivial proof. Note that Reddi et al. [24] used PL-SVRG/SAGA to restart ProxSVRG/SAGA $O(\log(1/\epsilon))$ times to obtain the linear convergence rate under PL condition.

Moreover, similar to Table 2, if we choose $b = 1$ or $n$ for ProxSVRG+, then its convergence result is $O(\frac{n}{\mu}\log\frac{1}{\epsilon})$, which is the same as ProxGD [13]. If we choose $b = n^{2/3}$ for ProxSVRG+, then the convergence result is $O(\frac{n^{2/3}}{\mu}\log\frac{1}{\epsilon})$, the same as the best result achieved by ProxSVRG/SAGA [24]. If we choose $b = 1/(\mu\epsilon)^{2/3}$ (assuming $1/(\mu\epsilon) < n$) for ProxSVRG+, then its convergence result is $O(\frac{1}{\mu^{5/3}\epsilon^{2/3}}\log\frac{1}{\epsilon})$ which generalizes the best result of SCSG [17] to the more general nonsmooth nonconvex case and is better than ProxGD and ProxSVRG/SAGA. Also note that our ProxSVRG+ uses much less proximal oracle calls than ProxSVRG/SAGA if $b < n^{2/3}$.

(2) Another benefit of ProxSVRG+ is that it can *automatically* switch to the faster linear convergence rate in some regions as long as the loss function satisfies the PL condition locally in these regions. This is impossible for ProxSVRG [24] since it needs to be restarted many times.

(3) We want to point out that [13, 24] used the following form of PL condition:

$$\exists \mu > 0, \text{ such that } D_h(x, \alpha) \geq 2\mu(\Phi(x) - \Phi^*), \ \forall x, \tag{8}$$

where $D_h(x, \alpha) := -2\alpha \min_y \big\{\langle\nabla f(x), y - x\rangle + \frac{\alpha}{2}\|y - x\|^2 + h(y) - h(x)\big\}$. Our PL condition is arguably more natural. In fact, one can show that if $\alpha = 1/\eta$, our new PL condition (7) implies (8). For a direct comparison with prior results, we also provide the proof of the same result of Theorem 2 using the previous PL condition (8) in the appendix.

The proofs of Theorem 2 under PL form (7) and (8) are provided in Appendix B.1 and B.2, respectively. Recently, Csiba and Richtárik [6] proposed a novel weakly PL condition. The (strongly) PL condition (7) or (8) serves as a generalization of strong convexity as we discussed in the beginning of this section. One can achieve linear convergence under (7) or (8). However, the weakly PL condition [6] may be considered as a generalization of (weak) convexity. Although one only achieves the sublinear convergence under this condition, it is still interesting to figure out similar (sublinear) convergence (for ProxSVRG+, ProxSVRG, etc.) under their weakly PL condition.

## 6 Experiments

In this section, we present the experimental results. We compare the nonconvex ProxSVRG+ with nonconvex ProxGD, ProxSGD [10], ProxSVRG [24]. We conduct the experiments using the non-negative principal component analysis (NN-PCA) problem (same as [24]). In general, NN-PCA is NP-hard. Specifically, the optimization problem for a given set of samples (i.e., $\{z_i\}_{i=1}^n$) is:

$$\min_{\|x\|\leq 1, x\geq 0} -\frac{1}{2}x^T\Big(\sum_{i=1}^n z_i z_i^T\Big)x. \tag{9}$$

Note that (9) can be written in the form (1), where $f(x) = \sum_{i=1}^n f_i(x) = \sum_{i=1}^n -\frac{1}{2}(x^T z_i)^2$ and $h(x) = I_C(x)$ where set $C = \{x \in \mathbb{R}^d | \|x\| \leq 1, x \geq 0\}$. We conduct the experiment on the standard MNIST and 'a9a' datasets. [6] The experimental results on both datasets (corresponding to the first row and second row in Figure 3–5) are almost the same.

The samples from each dataset are normalized, i.e., $\|z_i\| = 1$ for all $i \in n$. The parameters of the algorithms are chosen as follows: $L$ can be precomputed from the data samples $\{z_i\}_{i=1}^n$ in the same way as in [18]. The step sizes $\eta$ for different algorithms are set to be the ones used in their convergence results: For ProxGD, it is $\eta = 1/L$ (see Corollary 1 in [10]); for ProxSGD, $\eta = 1/(2L)$ (see Corollary 3 in [10]); for ProxSVRG, $\eta = b^{3/2}/(3Ln)$ (see Theorem 6 in [24]). The step size for our ProxSVRG+ is $1/(6L)$ (see our Theorem 1). We did not further tune the step sizes. The batch size $B$ (in Line 4 of Algorithm 1) is equal to $n/5$ (i.e., 20% data samples). We also considered $B = n/10$, the performance among these algorithms are similar to the case $B = n/5$. In practice, one can tune the step size $\eta$ and parameter $B$.

Regarding the comparison among these algorithms, we use the number of SFO calls (see Definition 2) to evaluate them since the number of PO calls of them are the same except ProxSVRG (which is already clearly demonstrated by Figure 2). Note that we amortize the batch size ($n$ or $B$ in Line 4 of Algorithm 1) into the inner loops, so that the curves in the figures are smoother, i.e., the number of SFO calls for each iteration (inner loop) of ProxSVRG and ProxSVRG+ is counted as $b + n/m$ and $b + B/m$, respectively. Note that ProxGD uses $n$ SFO calls (full gradient) in each iteration.

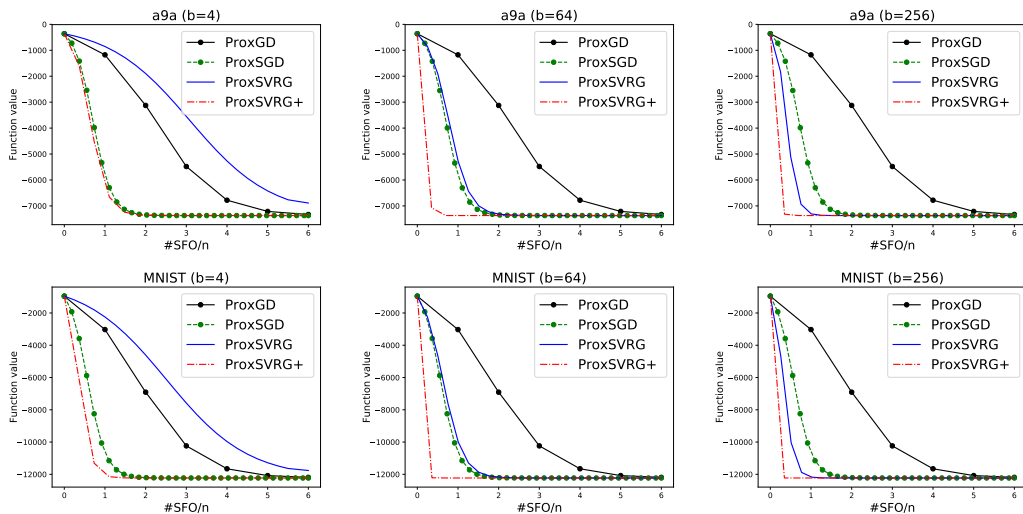

Figure 3: Comparison among algorithms with different minibatch size $b$

In Figure 3, we compare the performance of these four algorithms as we vary the minibatch size $b$. In particular, the first column ($b = 4$) shows that ProxSVRG+ and ProxSVRG perform similar to ProxSGD and ProxGD respectively, which is quite consistent with the theoretical results (Figure 1). Then, ProxSVRG+ and ProxSVRG both get better as $b$ increases. Note that our ProxSVRG+ performs better than ProxGD, ProxSGD and ProxSVRG.

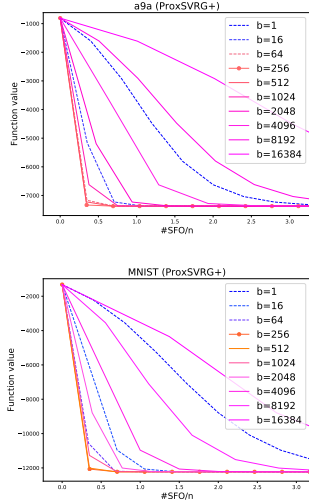
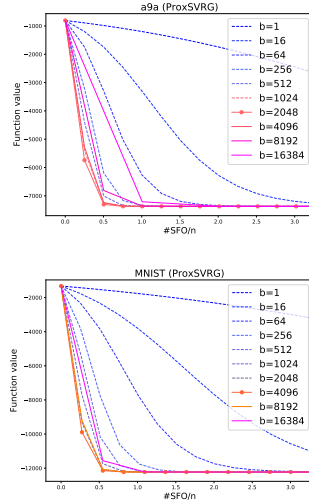
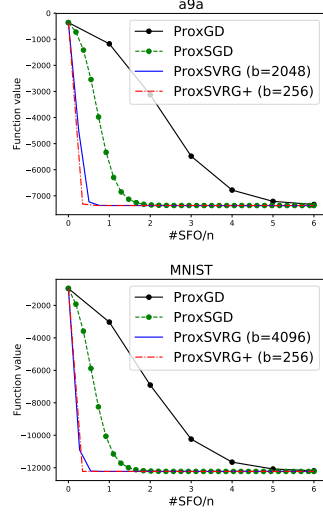

Figure 4: ProxSVRG+ and ProxSVRG under different $b$          Figure 5: Under the best $b$

Figure 4 demonstrates that our ProxSVRG+ prefers smaller minibatch sizes than ProxSVRG (see the curves with dots). Then, in Figure 5, we compare the algorithms with their corresponding best minibatch size $b$.

In conclusion, the experimental results are quite consistent with the theoretical results, i.e., different algorithms favor different minibatch sizes (see Figure 1). Concretely, our ProxSVRG+ achieves its best performance with a moderate minibatch size $b = 256$ unlike ProxSVRG with $b = 2048/4096$. Besides, choosing $b = 64$ is already good enough for ProxSVRG+ by comparing the second column and last column of Figure 3, however ProxSVRG is only as good as ProxSGD with such a minibatch size. Moreover, ProxSVRG+ uses much less proximal oracle calls than ProxSVRG if $b < n^{2/3}$ (see Figure 2). Note that small minibatch size also usually provides better generalization in practice. Thus, we argue that our ProxSVRG+ might be more attractive in certain applications due to its moderate minibatch size.

## 7 Conclusion

In this paper, we propose a simple proximal stochastic method called ProxSVRG+ for nonsmooth nonconvex optimization. We prove that ProxSVRG+ improves/generalizes several well-known convergence results (e.g., ProxGD, ProxSGD, ProxSVRG/SAGA and SCSG) by choosing proper minibatch sizes. In particular, ProxSVRG+ is $\sqrt{b}$ (or $\sqrt{b}\epsilon n$ if $n > 1/\epsilon$) times faster than ProxGD, which partially answers the open problem (i.e., developing stochastic methods with provably better performance than ProxGD with constant minibatch size $b$) proposed in [24]. Also, ProxSVRG+ generalizes the results of SCSG [17] to this nonsmooth nonconvex case, and it is more straightforward than SCSG and yields simpler analysis. Moreover, for nonconvex functions satisfying Polyak-Łojasiewicz condition, we prove that ProxSVRG+ achieves the global linear convergence rate without restart. As a result, ProxSVRG+ can *automatically* switch to the faster linear convergence rate (i.e., $O(\log 1/\epsilon)$) from sublinear convergence rate (i.e., $O(1/\epsilon)$) in some regions (e.g., the neighborhood of a local minimum) as long as the objective function satisfies the PL condition locally in these regions. This is impossible for ProxSVRG [24] since it needs to be restarted $O(\log 1/\epsilon)$ times.

### Acknowledgments

This research is supported in part by the National Basic Research Program of China Grant 2015CB358700, the National Natural Science Foundation of China Grant 61772297, 61632016, 61761146003, and a grant from Microsoft Research Asia. The authors would like to thank Rong Ge (Duke), Xiangliang Zhang (KAUST) and the anonymous reviewers for their useful suggestions.

## Footnotes

[1]In fact, some studies argued that smaller minibatch sizes in SGD are very useful for generalization (e.g., [14]). Although generalization is not the focus of the present paper, it provides further motivation for studying the moderate minibatch size regime.

[4]If $B = n$, ProxSVRG+ is almost the same as ProxSVRG (i.e., $g^s = \frac{1}{n}\sum_{j=1}^n \nabla f_j(\widetilde{x}^{s-1}) = \nabla f(\widetilde{x}^{s-1})$) except some detailed parameter settings (e.g., step-size, epoch length).

[5]A similar observation was also made in Natasha1.5 [2]. However, Natasha1.5 divides each epoch into multiple sub-epochs and randomly chooses the iteration point at the end of each sub-epoch. In our ProxSVRG+, the length of an epoch is deterministic and it directly uses the last iteration point at the end of each epoch.

[6]The datasets can be downloaded from `https://www.csie.ntu.edu.tw/~cjlin/libsvmtools/datasets/`

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
