[Supplementary Material · supplementary_material_full_version_proxsvrgplus_2018.pdf]

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

# A  Proofs for Nonconvex ProxSVRG+ Algorithm

In this appendix, we first provide the proof of Theorem 1 (Appendix A.1). Then we provide the proof for other choices of epoch length $m$ (Appendix A.2).

## A.1  Proof of Theorem 1

Before proving Theorem 1, we need a useful lemma for the proximal operator.

**Lemma 1** *Let $x^+ := \mathrm{prox}_{\eta h}(x - \eta v)$, then the following inequality holds:*

$$\Phi(x^+) \leq \Phi(z) + \langle \nabla f(x) - v, x^+ - z \rangle - \frac{1}{\eta}\langle x^+ - x, x^+ - z \rangle + \frac{L}{2}\|x^+ - x\|^2 + \frac{L}{2}\|z - x\|^2, \ \ \forall z \in \mathbb{R}^d. \tag{10}$$

*Proof:* First, we recall the proximal operator (see (5)):

$$\mathrm{prox}_{\eta h}(x - \eta v) := \arg\min_{y \in \mathbb{R}^d} \left( h(y) + \frac{1}{2\eta}\|y - x\|^2 + \langle v, y \rangle \right). \tag{11}$$

For the nonsmooth function $h(x)$, we have

$$h(x^+) \leq h(z) + \langle p, x^+ - z \rangle \tag{12}$$

$$= h(z) - \langle v + \frac{1}{\eta}(x^+ - x), x^+ - z \rangle, \tag{13}$$

where $p \in \partial h(x^+)$ such that $p + \frac{1}{\eta}(x^+ - x) + v = 0$ according to the optimality condition of (11), and (12) holds due to the convexity of $h$.

For the nonconvex function $f(x)$, we have

$$f(x^+) \leq f(x) + \langle \nabla f(x), x^+ - x \rangle + \frac{L}{2}\|x^+ - x\|^2 \tag{14}$$

$$-f(z) \leq -f(x) + \langle -\nabla f(x), z - x \rangle + \frac{L}{2}\|z - x\|^2, \tag{15}$$

where (14) holds since $f(x)$ has $L$-Lipschitz continuous gradient (see (2)), and (15) holds since $-f(x)$ has the same $L$-Lipschitz continuous gradient as $f(x)$.

This lemma is proved by adding (13), (14), (15), and recalling $\Phi(x) = f(x) + h(x)$.  □

**Proof of Theorem 1.** Now, we are ready to use Lemma 1 to prove Theorem 1. Let $x_t^s := \mathrm{prox}_{\eta h}(x_{t-1}^s - \eta v_{t-1}^s)$ and $\bar{x}_t^s := \mathrm{prox}_{\eta h}\big(x_{t-1}^s - \eta \nabla f(x_{t-1}^s)\big)$. We further define some notations for the frequently occurring items. Let $\Delta x_t^s := x_t^s - x_{t-1}^s$, $\Delta \bar{x}_t^s := \bar{x}_t^s - x_{t-1}^s$ and $\xi_{t-1}^s := \nabla f(x_{t-1}^s) - v_{t-1}^s$.

By letting $x^+ = x_t^s, x = x_{t-1}^s, v = v_{t-1}^s$ and $z = \bar{x}_t^s$ in (10), we have

$$\Phi(x_t^s) \leq \Phi(\bar{x}_t^s) + \langle \xi_{t-1}^s, x_t^s - \bar{x}_t^s \rangle - \frac{1}{\eta}\langle \Delta x_t^s, x_t^s - \bar{x}_t^s \rangle + \frac{L}{2}\|\Delta x_t^s\|^2 + \frac{L}{2}\|\Delta \bar{x}_t^s\|^2. \tag{16}$$

Besides, by letting $x^+ = \bar{x}_t^s, x = x_{t-1}^s, v = \nabla f(x_{t-1}^s)$ and $z = x = x_{t-1}^s$ in (10), we have

$$\Phi(\bar{x}_t^s) \leq \Phi(x_{t-1}^s) - \frac{1}{\eta}\langle \Delta \bar{x}_t^s, \Delta \bar{x}_t^s \rangle + \frac{L}{2}\|\Delta \bar{x}_t^s\|^2 = \Phi(x_{t-1}^s) - \left(\frac{1}{\eta} - \frac{L}{2}\right)\|\Delta \bar{x}_t^s\|^2. \tag{17}$$

We add (16) and (17) to obtain the key inequality

$$\Phi(x_t^s) \le \Phi(x_{t-1}^s) + \langle \xi_{t-1}^s, x_t^s - \bar{x}_t^s \rangle - \frac{1}{\eta}\langle \Delta x_t^s, x_t^s - \bar{x}_t^s \rangle + \frac{L}{2}\|\Delta x_t^s\|^2 - \Big(\frac{1}{\eta} - L\Big)\|\Delta \bar{x}_t^s\|^2$$

$$= \Phi(x_{t-1}^s) + \langle \xi_{t-1}^s, x_t^s - \bar{x}_t^s \rangle + \frac{L}{2}\|\Delta x_t^s\|^2 - \Big(\frac{1}{\eta} - L\Big)\|\Delta \bar{x}_t^s\|^2$$

$$\quad - \frac{1}{2\eta}\big(\|\Delta x_t^s\|^2 + \|x_t^s - \bar{x}_t^s\|^2 - \|\Delta \bar{x}_t^s\|^2\big)$$

$$= \Phi(x_{t-1}^s) - \Big(\frac{1}{2\eta} - \frac{L}{2}\Big)\|\Delta x_t^s\|^2 - \Big(\frac{1}{2\eta} - L\Big)\|\Delta \bar{x}_t^s\|^2 + \langle \xi_{t-1}^s, x_t^s - \bar{x}_t^s \rangle - \frac{1}{2\eta}\|x_t^s - \bar{x}_t^s\|^2$$

$$\le \Phi(x_{t-1}^s) - \Big(\frac{1}{2\eta} - \frac{L}{2}\Big)\|\Delta x_t^s\|^2 - \Big(\frac{1}{2\eta} - L\Big)\|\Delta \bar{x}_t^s\|^2 + \langle \xi_{t-1}^s, x_t^s - \bar{x}_t^s \rangle$$

$$\quad - \frac{1}{8\eta}\|\Delta x_t^s\|^2 + \frac{1}{6\eta}\|\Delta \bar{x}_t^s\|^2 \tag{18}$$

$$= \Phi(x_{t-1}^s) - \Big(\frac{5}{8\eta} - \frac{L}{2}\Big)\|\Delta x_t^s\|^2 - \Big(\frac{1}{3\eta} - L\Big)\|\Delta \bar{x}_t^s\|^2 + \langle \xi_{t-1}^s, x_t^s - \bar{x}_t^s \rangle$$

$$\le \Phi(x_{t-1}^s) - \Big(\frac{5}{8\eta} - \frac{L}{2}\Big)\|\Delta x_t^s\|^2 - \Big(\frac{1}{3\eta} - L\Big)\|\Delta \bar{x}_t^s\|^2 + \eta\|\xi_{t-1}^s\|^2, \tag{19}$$

where (18) uses the following Young's inequality (choose $\alpha = 3$)

$$\|x_t^s - x_{t-1}^s\|^2 \le \Big(1 + \frac{1}{\alpha}\Big)\|\bar{x}_t^s - x_{t-1}^s\|^2 + (1 + \alpha)\|x_t^s - \bar{x}_t^s\|^2, \quad \forall \alpha > 0, \tag{20}$$

and (19) holds due to the following Lemma 2.

**Lemma 2** *Let $x_t^s := \mathrm{prox}_{\eta h}(x_{t-1}^s - \eta v_{t-1}^s)$ and $\bar{x}_t^s := \mathrm{prox}_{\eta h}\big(x_{t-1}^s - \eta \nabla f(x_{t-1}^s)\big)$. Then, the following inequality holds:*

$$\langle \xi_{t-1}^s, x_t^s - \bar{x}_t^s \rangle \le \eta\|\xi_{t-1}^s\|^2$$

**Proof of Lemma 2.** First, we obtain the relation between $\|x_t^s - \bar{x}_t^s\|$ and $\|\xi_{t-1}^s\|$ as follows (similar to [10]):

$$h(x_t^s) \le h(\bar{x}_t^s) - \langle v_{t-1}^s + \frac{1}{\eta}(x_t^s - x_{t-1}^s), x_t^s - \bar{x}_t^s \rangle \tag{21}$$

$$h(\bar{x}_t^s) \le h(x_t^s) - \langle \nabla f(x_{t-1}^s) + \frac{1}{\eta}(\bar{x}_t^s - x_{t-1}^s), \bar{x}_t^s - x_t^s \rangle, \tag{22}$$

where (21) and (22) hold due to (13). Adding (21) and (22), we have

$$\frac{1}{\eta}\langle x_t^s - \bar{x}_t^s, x_t^s - \bar{x}_t^s \rangle \le \langle \nabla f(x_{t-1}^s) - v_{t-1}^s, x_t^s - \bar{x}_t^s \rangle$$

$$\frac{1}{\eta}\|x_t^s - \bar{x}_t^s\|^2 \le \|\nabla f(x_{t-1}^s) - v_{t-1}^s\|\|x_t^s - \bar{x}_t^s\| \tag{23}$$

$$\|x_t^s - \bar{x}_t^s\| \le \eta\|\nabla f(x_{t-1}^s) - v_{t-1}^s\| = \eta\|\xi_{t-1}^s\|, \tag{24}$$

where (23) uses Cauchy-Schwarz inequality.

Now, this lemma is proved by using Cauchy-Schwarz inequality and (24), i.e., $\langle \xi_{t-1}^s, x_t^s - \bar{x}_t^s \rangle \le \|\xi_{t-1}^s\|\|x_t^s - \bar{x}_t^s\| \le \eta\|\xi_{t-1}^s\|^2$. $\qquad\square$

Note that $x_t^s = \mathrm{prox}_{\eta h}(x_{t-1}^s - \eta v_{t-1}^s)$ is the iterated form in our algorithm (see Line 7 in Algorithm 1). Now, we take expectations with all history for (19).

$$\mathbb{E}[\Phi(x_t^s)] \le \mathbb{E}\Big[\Phi(x_{t-1}^s) - \Big(\frac{5}{8\eta} - \frac{L}{2}\Big)\|\Delta x_t^s\|^2 - \Big(\frac{1}{3\eta} - L\Big)\|\Delta \bar{x}_t^s\|^2 + \eta\|\xi_{t-1}^s\|^2\Big] \tag{25}$$

Then, we bound the variance term in (25) as follows:

$$\mathbb{E}\Big[\eta\|\xi_{t-1}^s\|^2\Big]$$

$$= \mathbb{E}\Big[\eta\Big\|\frac{1}{b}\sum_{i\in I_b}\Big(\nabla f_i(x_{t-1}^s) - \nabla f_i(\widetilde{x}^{s-1})\Big) - \Big(\nabla f(x_{t-1}^s) - g^s\Big)\Big\|^2\Big]$$

$$= \mathbb{E}\Big[\eta\Big\|\frac{1}{b}\sum_{i\in I_b}\Big(\nabla f_i(x_{t-1}^s) - \nabla f_i(\widetilde{x}^{s-1})\Big) - \Big(\nabla f(x_{t-1}^s) - \frac{1}{B}\sum_{j\in I_B}\nabla f_j(\widetilde{x}^{s-1})\Big)\Big\|^2\Big]$$

$$= \mathbb{E}\Big[\eta\Big\|\frac{1}{b}\sum_{i\in I_b}\Big(\nabla f_i(x_{t-1}^s) - \nabla f_i(\widetilde{x}^{s-1})\Big) - \Big(\nabla f(x_{t-1}^s) - \nabla f(\widetilde{x}^{s-1})\Big)$$
$$\qquad\qquad + \Big(\frac{1}{B}\sum_{j\in I_B}\nabla f_j(\widetilde{x}^{s-1}) - \nabla f(\widetilde{x}^{s-1})\Big)\Big\|^2\Big]$$

$$= \eta\mathbb{E}\Big[\Big\|\frac{1}{b}\sum_{i\in I_b}\Big((\nabla f_i(x_{t-1}^s) - \nabla f_i(\widetilde{x}^{s-1})) - (\nabla f(x_{t-1}^s) - \nabla f(\widetilde{x}^{s-1}))\Big)$$
$$\qquad\qquad + \frac{1}{B}\sum_{j\in I_B}\Big(\nabla f_j(\widetilde{x}^{s-1}) - \nabla f(\widetilde{x}^{s-1})\Big)\Big\|^2\Big]$$

$$= \eta\mathbb{E}\Big[\Big\|\frac{1}{b}\sum_{i\in I_b}\Big((\nabla f_i(x_{t-1}^s) - \nabla f_i(\widetilde{x}^{s-1})) - (\nabla f(x_{t-1}^s) - \nabla f(\widetilde{x}^{s-1}))\Big)\Big\|^2\Big]$$
$$\qquad\qquad + \eta\mathbb{E}\Big[\Big\|\frac{1}{B}\sum_{j\in I_B}\Big(\nabla f_j(\widetilde{x}^{s-1}) - \nabla f(\widetilde{x}^{s-1})\Big)\Big\|^2\Big] \qquad (26)$$

$$= \frac{\eta}{b^2}\mathbb{E}\Big[\sum_{i\in I_b}\Big\|\Big((\nabla f_i(x_{t-1}^s) - \nabla f_i(\widetilde{x}^{s-1})) - (\nabla f(x_{t-1}^s) - \nabla f(\widetilde{x}^{s-1}))\Big)\Big\|^2\Big]$$
$$\qquad\qquad + \eta\mathbb{E}\Big[\Big\|\frac{1}{B}\sum_{j\in I_B}\Big(\nabla f_j(\widetilde{x}^{s-1}) - \nabla f(\widetilde{x}^{s-1})\Big)\Big\|^2\Big] \qquad (27)$$

$$\leq \frac{\eta}{b^2}\mathbb{E}\Big[\sum_{i\in I_b}\|\nabla f_i(x_{t-1}^s) - \nabla f_i(\widetilde{x}^{s-1})\|^2\Big] + \eta\mathbb{E}\Big[\Big\|\frac{1}{B}\sum_{j\in I_B}\Big(\nabla f_j(\widetilde{x}^{s-1}) - \nabla f(\widetilde{x}^{s-1})\Big)\Big\|^2\Big] \qquad (28)$$

$$\leq \frac{\eta L^2}{b}\mathbb{E}[\|x_{t-1}^s - \widetilde{x}^{s-1}\|^2] + \frac{I\{B<n\}\eta\sigma^2}{B}, \qquad (29)$$

where the expectations are taking with $I_b$ and $I_B$. (26) and (27) hold since $\mathbb{E}[\|x_1 + x_2 + \cdots + x_k\|^2] = \sum_{i=1}^k \mathbb{E}[\|x_i\|^2]$ if $x_1, x_2, \ldots, x_k$ are independent and of mean zero (note that $I_b$ and $I_B$ are also independent). (28) uses the fact that $\mathbb{E}[\|x - \mathbb{E}[x]\|^2] \leq \mathbb{E}[\|x\|^2]$, for any random variable $x$. (29) holds due to (2) and Assumption 1.

Now, we plug (29) into (25) to obtain (where $D := \frac{I\{B<n\}\eta\sigma^2}{B}$)

$$\mathbb{E}[\Phi(x_t^s)] \leq \mathbb{E}\Big[\Phi(x_{t-1}^s) - \Big(\frac{5}{8\eta} - \frac{L}{2}\Big)\|\Delta x_t^s\|^2 - \Big(\frac{1}{3\eta} - L\Big)\|\Delta \bar{x}_t^s\|^2 + \frac{\eta L^2}{b}\|x_{t-1}^s - \widetilde{x}^{s-1}\|^2 + D\Big] \qquad (30)$$

$$= \mathbb{E}\Big[\Phi(x_{t-1}^s) - \frac{13L}{4}\|\Delta x_t^s\|^2 - L\|\Delta \bar{x}_t^s\|^2 + \frac{L}{6b}\|x_{t-1}^s - \widetilde{x}^{s-1}\|^2 + D\Big] \qquad (31)$$

$$= \mathbb{E}\Big[\Phi(x_{t-1}^s) - \frac{13L}{4}\|\Delta x_t^s\|^2 - \frac{1}{36L}\|\mathcal{G}_\eta(x_{t-1}^s)\|^2 + \frac{L}{6b}\|x_{t-1}^s - \widetilde{x}^{s-1}\|^2 + D\Big] \qquad (32)$$

$$\leq \mathbb{E}\Big[\Phi(x_{t-1}^s) - \frac{13L}{8t}\|x_t^s - \widetilde{x}^{s-1}\|^2 - \frac{1}{36L}\|\mathcal{G}_\eta(x_{t-1}^s)\|^2$$
$$\qquad\qquad + \Big(\frac{L}{6b} + \frac{13L}{8t-4}\Big)\|x_{t-1}^s - \widetilde{x}^{s-1}\|^2 + D\Big], \qquad (33)$$

where (31) uses $\eta = \frac{1}{6L}$, and (32) uses the definition of gradient mapping $\mathcal{G}_\eta(x_{t-1}^s)$ (see (4)) and recall $\bar{x}_t^s := \text{prox}_{\eta h}\big(x_{t-1}^s - \eta\nabla f(x_{t-1}^s)\big)$. (33) uses $\|x_t^s - \tilde{x}^{s-1}\|^2 \leq \big(1 + \frac{1}{\alpha}\big)\|x_{t-1}^s - \tilde{x}^{s-1}\|^2 + (1+\alpha)\|x_t^s - x_{t-1}^s\|^2$ by choosing $\alpha = 2t - 1$.

Now, adding (33) for all iterations $1 \leq t \leq m$ in epoch $s$ and recalling that $x_m^s = \tilde{x}^s$ and $x_0^s = \tilde{x}^{s-1}$, we get

$$\mathbb{E}[\Phi(\tilde{x}^s)]$$

$$\leq \mathbb{E}\Big[\Phi(\tilde{x}^{s-1}) - \sum_{t=1}^m \frac{1}{36L}\|\mathcal{G}_\eta(x_{t-1}^s)\|^2 - \sum_{t=1}^m \frac{13L}{8t}\|x_t^s - \tilde{x}^{s-1}\|^2$$

$$+ \sum_{t=1}^m \Big(\frac{L}{6b} + \frac{13L}{8t-4}\Big)\|x_{t-1}^s - \tilde{x}^{s-1}\|^2 + \sum_{t=1}^m D\Big]$$

$$\leq \mathbb{E}\Big[\Phi(\tilde{x}^{s-1}) - \sum_{t=1}^m \frac{1}{36L}\|\mathcal{G}_\eta(x_{t-1}^s)\|^2 - \sum_{t=1}^{m-1} \frac{13L}{8t}\|x_t^s - \tilde{x}^{s-1}\|^2$$

$$+ \sum_{t=2}^m \Big(\frac{L}{6b} + \frac{13L}{8t-4}\Big)\|x_{t-1}^s - \tilde{x}^{s-1}\|^2 + \sum_{t=1}^m D\Big] \tag{34}$$

$$= \mathbb{E}\Big[\Phi(\tilde{x}^{s-1}) - \sum_{t=1}^m \frac{1}{36L}\|\mathcal{G}_\eta(x_{t-1}^s)\|^2 - \sum_{t=1}^{m-1} \Big(\frac{13L}{8t} - \frac{L}{6b} - \frac{13L}{8t+4}\Big)\|x_t^s - \tilde{x}^{s-1}\|^2 + \sum_{t=1}^m D\Big]$$

$$\leq \mathbb{E}\Big[\Phi(\tilde{x}^{s-1}) - \sum_{t=1}^m \frac{1}{36L}\|\mathcal{G}_\eta(x_{t-1}^s)\|^2 - \sum_{t=1}^{m-1} \Big(\frac{L}{2t^2} - \frac{L}{6b}\Big)\|x_t^s - \tilde{x}^{s-1}\|^2 + \sum_{t=1}^m D\Big]$$

$$\leq \mathbb{E}\Big[\Phi(\tilde{x}^{s-1}) - \sum_{t=1}^m \frac{1}{36L}\|\mathcal{G}_\eta(x_{t-1}^s)\|^2 + \sum_{t=1}^m D\Big], \tag{35}$$

where (34) holds since $\|\cdot\|^2$ always be non-negative and $x_0^s = \tilde{x}^{s-1}$, and (35) holds since $m = \sqrt{b}$. Thus, $\frac{L}{2t^2} - \frac{L}{6b} \geq 0$ for all $1 \leq t < m$.

Now, we sum up (35) for all epochs $1 \leq s \leq S$ to finish the proof as follows:

$$0 \leq \mathbb{E}[\Phi(\tilde{x}^S) - \Phi(x^*)] \leq \mathbb{E}\Big[\Phi(\tilde{x}^0) - \Phi(x^*) - \sum_{s=1}^S\sum_{t=1}^m \frac{1}{36L}\|\mathcal{G}_\eta(x_{t-1}^s)\|^2 + \sum_{s=1}^S\sum_{t=1}^m D\Big]$$

$$\mathbb{E}[\|\mathcal{G}_\eta(\hat{x})\|^2] \leq \frac{36L\big(\Phi(x_0) - \Phi(x^*)\big)}{Sm} + \frac{I\{B < n\}36L\eta\sigma^2}{B} \tag{36}$$

$$= \frac{36L\big(\Phi(x_0) - \Phi(x^*)\big)}{Sm} + \frac{I\{B < n\}6\sigma^2}{B} = 2\epsilon, \tag{37}$$

where (36) holds since $\hat{x}$ is chosen uniformly randomly from $\{x_{t-1}^s\}_{t\in[m],s\in[S]}$ and recall $D = \frac{I\{B<n\}\eta\sigma^2}{B}$, and (37) uses $\eta = \frac{1}{6L}$. Now, we obtain the total number of iterations $T = Sm = S\sqrt{b} = \frac{36L\big(\Phi(x_0)-\Phi(x^*)\big)}{\epsilon}$. The number of PO calls equals to $T = Sm = \frac{36L\big(\Phi(x_0)-\Phi(x^*)\big)}{\epsilon}$. The proof is finished since the number of SFO calls equals to $Sn + Smb = 36L\big(\Phi(x_0) - \Phi(x^*)\big)\big(\frac{n}{\epsilon\sqrt{b}} + \frac{b}{\epsilon}\big)$ if $B = n$ (i.e., the second term in (37) is 0 and thus Assumption 1 is not needed), or equals to $SB + Smb = 36L\big(\Phi(x_0) - \Phi(x^*)\big)\big(\frac{B}{\epsilon\sqrt{b}} + \frac{b}{\epsilon}\big)$ if $B < n$ (note that $\frac{I\{B<n\}6\sigma^2}{B} \leq \epsilon$ since $B \geq 6\sigma^2/\epsilon$). $\qquad\square$

## A.2 Other Choices of Epoch Length $m$

In this section, we show that the similar convergence result (i.e., Theorem 1) holds for other choices of epoch length $m \neq \sqrt{b}$. The difference is that we need to choose different step size $\eta$. Now, we list the similar convergence result in the following theorem and then prove it.

**Theorem 3** *Let step size $\eta = \min\{\frac{1}{6L}, \frac{\sqrt{b}}{6mL}\}$, where $b$ is the minibatch size and $m$ is the epoch length. Then $\hat{x}$ returned by Algorithm 1 is an $\epsilon$-accurate solution for problem (1) (i.e., $\mathbb{E}[\|\mathcal{G}_\eta(\hat{x})\|^2] \leq \epsilon$). We distinguish the following two cases:*

*1) We let batch size $B = n$. The number of SFO calls is at most*

$$6\big(\Phi(x_0) - \Phi(x^*)\big)\Big(\frac{n}{\epsilon\eta m} + \frac{b}{\epsilon\eta}\Big).$$

*2) Under Assumption 1, we let batch size $B = \min\{6\sigma^2/\epsilon, n\}$. The number of SFO calls is at most*

$$6\big(\Phi(x_0) - \Phi(x^*)\big)\Big(\frac{B}{\epsilon\eta m} + \frac{b}{\epsilon\eta}\Big).$$

*In both cases, the number of PO calls equals to the total number of iterations $T$ which is at most $\frac{6\big(\Phi(x_0) - \Phi(x^*)\big)}{\epsilon\eta}$.*

*Proof:* First, we recall the Inequality (30) in the proof of Theorem 1 as follows (recall that $\Delta x_t^s := x_t^s - x_{t-1}^s$, $\Delta \bar{x}_t^s := \bar{x}_t^s - x_{t-1}^s$ and $D := \frac{I\{B<n\}\eta\sigma^2}{B}$):

$$\mathbb{E}[\Phi(x_t^s)]$$
$$\leq \mathbb{E}\Big[\Phi(x_{t-1}^s) - \Big(\frac{5}{8\eta} - \frac{L}{2}\Big)\|\Delta x_t^s\|^2 - \Big(\frac{1}{3\eta} - L\Big)\|\Delta \bar{x}_t^s\|^2 + \frac{\eta L^2}{b}\|x_{t-1}^s - \tilde{x}^{s-1}\|^2 + D\Big]$$
$$= \mathbb{E}\Big[\Phi(x_{t-1}^s) - \Big(\frac{5}{8\eta} - \frac{L}{2}\Big)\|\Delta x_t^s\|^2 - \Big(\frac{1}{3\eta} - L\Big)\eta^2\|\mathcal{G}_\eta(x_{t-1}^s)\|^2 + \frac{\eta L^2}{b}\|x_{t-1}^s - \tilde{x}^{s-1}\|^2 + D\Big]$$
$$\tag{38}$$
$$\leq \mathbb{E}\Big[\Phi(x_{t-1}^s) - \Big(\frac{5}{8\eta} - \frac{L}{2}\Big)\|\Delta x_t^s\|^2 - \frac{\eta}{6}\|\mathcal{G}_\eta(x_{t-1}^s)\|^2 + \frac{\eta L^2}{b}\|x_{t-1}^s - \tilde{x}^{s-1}\|^2 + D\Big] \tag{39}$$
$$\leq \mathbb{E}\Big[\Phi(x_{t-1}^s) - \frac{1}{2t}\Big(\frac{5}{8\eta} - \frac{L}{2}\Big)\|x_t^s - \tilde{x}^{s-1}\|^2 - \frac{\eta}{6}\|\mathcal{G}_\eta(x_{t-1}^s)\|^2$$
$$+ \Big(\frac{\eta L^2}{b} + \frac{1}{2t-1}\Big(\frac{5}{8\eta} - \frac{L}{2}\Big)\Big)\|x_{t-1}^s - \tilde{x}^{s-1}\|^2 + D\Big], \tag{40}$$

where (38) uses the definition of gradient mapping $\mathcal{G}_\eta(x_{t-1}^s)$ (see (4)) and recall $\bar{x}_t^s := \text{prox}_{\eta h}\big(x_{t-1}^s - \eta\nabla f(x_{t-1}^s)\big)$. (39) uses $\eta \leq \frac{1}{6L}$. (40) uses $\|x_t^s - \tilde{x}^{s-1}\|^2 \leq \big(1 + \frac{1}{\alpha}\big)\|x_{t-1}^s - \tilde{x}^{s-1}\|^2 + (1+\alpha)\|x_t^s - x_{t-1}^s\|^2$ by choosing $\alpha = 2t - 1$.

Now, the remaining proof is almost the same as that of Theorem 1. Adding $(40)$ for all iterations $1 \le t \le m$ in epoch $s$ and recalling that $x_m^s = \widetilde{x}^s$ and $x_0^s = \widetilde{x}^{s-1}$, we have

$$\mathbb{E}[\Phi(\widetilde{x}^s)]$$

$$\le \mathbb{E}\Big[\Phi(\widetilde{x}^{s-1}) - \sum_{t=1}^{m}\frac{\eta}{6}\|\mathcal{G}_\eta(x_{t-1}^s)\|^2 - \sum_{t=1}^{m}\frac{1}{2t}\Big(\frac{5}{8\eta}-\frac{L}{2}\Big)\|x_t^s - \widetilde{x}^{s-1}\|^2$$

$$+ \sum_{t=1}^{m}\Big(\frac{\eta L^2}{b}+\frac{1}{2t-1}\Big(\frac{5}{8\eta}-\frac{L}{2}\Big)\Big)\|x_{t-1}^s - \widetilde{x}^{s-1}\|^2 + \sum_{t=1}^{m}D\Big]$$

$$\le \mathbb{E}\Big[\Phi(\widetilde{x}^{s-1}) - \sum_{t=1}^{m}\frac{\eta}{6}\|\mathcal{G}_\eta(x_{t-1}^s)\|^2 - \sum_{t=1}^{m-1}\frac{1}{2t}\Big(\frac{5}{8\eta}-\frac{L}{2}\Big)\|x_t^s - \widetilde{x}^{s-1}\|^2$$

$$+ \sum_{t=2}^{m}\Big(\frac{\eta L^2}{b}+\frac{1}{2t-1}\Big(\frac{5}{8\eta}-\frac{L}{2}\Big)\Big)\|x_{t-1}^s - \widetilde{x}^{s-1}\|^2 + \sum_{t=1}^{m}D\Big] \qquad (41)$$

$$= \mathbb{E}\Big[\Phi(\widetilde{x}^{s-1}) - \sum_{t=1}^{m}\frac{\eta}{6}\|\mathcal{G}_\eta(x_{t-1}^s)\|^2 - \sum_{t=1}^{m-1}\Big(\Big(\frac{1}{2t}-\frac{1}{2t+1}\Big)\Big(\frac{5}{8\eta}-\frac{L}{2}\Big)-\frac{\eta L^2}{b}\Big)\|x_t^s - \widetilde{x}^{s-1}\|^2$$

$$+ \sum_{t=1}^{m}D\Big]$$

$$\le \mathbb{E}\Big[\Phi(\widetilde{x}^{s-1}) - \sum_{t=1}^{m}\frac{\eta}{6}\|\mathcal{G}_\eta(x_{t-1}^s)\|^2 - \sum_{t=1}^{m-1}\Big(\frac{1}{6t^2}\Big(\frac{5}{8\eta}-\frac{L}{2}\Big)-\frac{\eta L^2}{b}\Big)\|x_t^s - \widetilde{x}^{s-1}\|^2 + \sum_{t=1}^{m}D\Big]$$

$$\le \mathbb{E}\Big[\Phi(\widetilde{x}^{s-1}) - \sum_{t=1}^{m}\frac{\eta}{6}\|\mathcal{G}_\eta(x_{t-1}^s)\|^2 + \sum_{t=1}^{m}D\Big], \qquad (42)$$

where $(41)$ holds since $\|\cdot\|^2$ always be non-negative and $x_0^s = \widetilde{x}^{s-1}$, and $(42)$ holds since it is sufficient to show that $\big(\frac{1}{6m^2}\big(\frac{5}{8\eta}-\frac{L}{2}\big)-\frac{\eta L^2}{b}\big) \ge 0$. This holds since $\eta = \min\{\frac{1}{6L},\frac{\sqrt{b}}{6mL}\}$.

Now, we sum up $(42)$ for all epochs $1 \le s \le S$ to finish the proof as follows (recall $D := \frac{I\{B<n\}\eta\sigma^2}{B}$):

$$0 \le \mathbb{E}[\Phi(\widetilde{x}^S) - \Phi(x^*)] \le \mathbb{E}\Big[\Phi(\widetilde{x}^0) - \Phi(x^*) - \sum_{s=1}^{S}\sum_{t=1}^{m}\frac{\eta}{6}\|\mathcal{G}_\eta(x_{t-1}^s)\|^2 + \sum_{s=1}^{S}\sum_{t=1}^{m}\frac{I\{B<n\}\eta\sigma^2}{B}\Big]$$

$$\mathbb{E}[\|\mathcal{G}_\eta(\hat{x})\|^2] \le \frac{6\big(\Phi(x_0)-\Phi(x^*)\big)}{\eta Sm} + \frac{I\{B<n\}6\sigma^2}{B} = 2\epsilon, \qquad (43)$$

where $(43)$ holds since $\hat{x}$ is chosen uniformly randomly from $\{x_{t-1}^s\}_{t\in[m],s\in[S]}$. Now, we obtain the total number of iterations $T = Sm = \frac{6\big(\Phi(x_0)-\Phi(x^*)\big)}{\epsilon\eta}$. The number of PO calls equals to $T = Sm = \frac{6\big(\Phi(x_0)-\Phi(x^*)\big)}{\epsilon\eta}$. The proof is finished since the number of SFO calls equals to $Sn + Smb = 6\big(\Phi(x_0)-\Phi(x^*)\big)\big(\frac{n}{\epsilon\eta m}+\frac{b}{\epsilon\eta}\big)$ if $B = n$ (i.e., the second term in $(43)$ is 0 and thus Assumption 1 is not needed), or equals to $SB + Smb = 6\big(\Phi(x_0)-\Phi(x^*)\big)\big(\frac{B}{\epsilon\eta m}+\frac{b}{\epsilon\eta}\big)$ if $B < n$ (note that $\frac{I\{B<n\}6\sigma^2}{B} \le \epsilon$ since $B \ge 6\sigma^2/\epsilon$). $\qquad\square$

# B Proofs for ProxSVRG+ Under PL Condition

In this appendix, we first provide the proof of Theorem 2 under the PL condition with form (7) (Appendix B.1). Then we also provide the proof of Theorem 2 under the PL condition with form (8) (Appendix B.2).

## B.1 Proof Under PL Form (7)

**Proof of Theorem 2.** First, we recall a key Inequality (33) from the proof of Theorem 1, i.e.,

$$\mathbb{E}[\Phi(x_t^s)] \leq \mathbb{E}\Big[\Phi(x_{t-1}^s) - \frac{13L}{8t}\|x_t^s - \widetilde{x}^{s-1}\|^2 - \frac{1}{36L}\|\mathcal{G}_\eta(x_{t-1}^s)\|^2$$
$$+ \Big(\frac{L}{6b} + \frac{13L}{8t-4}\Big)\|x_{t-1}^s - \widetilde{x}^{s-1}\|^2 + D\Big]. \tag{44}$$

Recall that $D := \frac{I\{B<n\}\eta\sigma^2}{B}$. Then, we plug the following PL inequality (see (7))

$$\|\mathcal{G}_\eta(x_{t-1}^s)\|^2 \geq 2\mu(\Phi(x_{t-1}^s) - \Phi^*)$$

into (44) to get

$$\mathbb{E}[\Phi(x_t^s)] \leq \mathbb{E}\Big[\Phi(x_{t-1}^s) - \frac{13L}{8t}\|x_t^s - \widetilde{x}^{s-1}\|^2 - \frac{\mu}{18L}(\Phi(x_{t-1}^s) - \Phi^*)$$
$$+ \Big(\frac{L}{6b} + \frac{13L}{8t-4}\Big)\|x_{t-1}^s - \widetilde{x}^{s-1}\|^2 + D\Big].$$

Then, we obtain

$$\mathbb{E}[\Phi(x_t^s) - \Phi^*] \leq \mathbb{E}\Big[\Big(1 - \frac{\mu}{18L}\Big)\big(\Phi(x_{t-1}^s) - \Phi^*\big) - \frac{13L}{8t}\|x_t^s - \widetilde{x}^{s-1}\|^2$$
$$+ \Big(\frac{L}{6b} + \frac{13L}{8t-4}\Big)\|x_{t-1}^s - \widetilde{x}^{s-1}\|^2 + D\Big]. \tag{45}$$

Let $\alpha := 1 - \frac{\mu}{18L}$ and $\Psi_t^s := \frac{\mathbb{E}[\Phi(x_t^s)-\Phi^*]}{\alpha^t}$. Plugging them into (45), we have

$$\Psi_t^s \leq \Psi_{t-1}^s - \mathbb{E}\Big[\frac{13L}{8t\alpha^t}\|x_t^s - \widetilde{x}^{s-1}\|^2 - \frac{1}{\alpha^t}\Big(\frac{L}{6b} + \frac{13L}{8t-4}\Big)\|x_{t-1}^s - \widetilde{x}^{s-1}\|^2 - \frac{1}{\alpha^t}D\Big]. \tag{46}$$

Now, adding (46) from all iterations $1 \leq t \leq m$ in epoch $s$ and recalling that $x_m^s = \widetilde{x}^s$ and $x_0^s = \widetilde{x}^{s-1}$, we have

$$\mathbb{E}[\Phi(\widetilde{x}^s) - \Phi^*]$$

$$\leq \alpha^m \mathbb{E}[\Phi(\widetilde{x}^{s-1}) - \Phi^*] + \alpha^m \sum_{t=1}^m \frac{1}{\alpha^t} D$$

$$- \alpha^m \mathbb{E}\Big[\sum_{t=1}^m \frac{13L}{8t\alpha^t}\|x_t^s - \widetilde{x}^{s-1}\|^2 - \sum_{t=1}^m \frac{1}{\alpha^t}\Big(\frac{L}{6b} + \frac{13L}{8t-4}\Big)\|x_{t-1}^s - \widetilde{x}^{s-1}\|^2\Big]$$

$$= \alpha^m \mathbb{E}[\Phi(\widetilde{x}^{s-1}) - \Phi^*] + \frac{1-\alpha^m}{1-\alpha}D$$

$$- \alpha^m \mathbb{E}\Big[\sum_{t=1}^m \frac{13L}{8t\alpha^t}\|x_t^s - \widetilde{x}^{s-1}\|^2 - \sum_{t=1}^m \frac{1}{\alpha^t}\Big(\frac{L}{6b} + \frac{13L}{8t-4}\Big)\|x_{t-1}^s - \widetilde{x}^{s-1}\|^2\Big]$$

$$\leq \alpha^m \mathbb{E}[\Phi(\widetilde{x}^{s-1}) - \Phi^*] + \frac{1-\alpha^m}{1-\alpha}D$$

$$- \alpha^m \mathbb{E}\Big[\sum_{t=1}^{m-1} \frac{13L}{8t\alpha^t}\|x_t^s - \widetilde{x}^{s-1}\|^2 - \sum_{t=2}^m \frac{1}{\alpha^t}\Big(\frac{L}{6b} + \frac{13L}{8t-4}\Big)\|x_{t-1}^s - \widetilde{x}^{s-1}\|^2\Big] \qquad (47)$$

$$= \alpha^m \mathbb{E}[\Phi(\widetilde{x}^{s-1}) - \Phi^*] + \frac{1-\alpha^m}{1-\alpha}D$$

$$- \alpha^m \mathbb{E}\Big[\sum_{t=1}^{m-1} \frac{1}{\alpha^{t+1}}\Big(\frac{13L\alpha}{8t} - \frac{L}{6b} - \frac{13L}{8t+4}\Big)\|x_t^s - \widetilde{x}^{s-1}\|^2\Big]$$

$$\leq \alpha^m \mathbb{E}[\Phi(\widetilde{x}^{s-1}) - \Phi^*] + \frac{1-\alpha^m}{1-\alpha}D$$

$$- \alpha^m \mathbb{E}\Big[\sum_{t=1}^{m-1} \frac{1}{\alpha^{t+1}}\Big(\frac{13L}{8t}\Big(1 - \frac{1}{18\sqrt{n}}\Big) - \frac{L}{6b} - \frac{13L}{8t+4}\Big)\|x_t^s - \widetilde{x}^{s-1}\|^2\Big] \qquad (48)$$

$$\leq \alpha^m \mathbb{E}[\Phi(\widetilde{x}^{s-1}) - \Phi^*] + \frac{1-\alpha^m}{1-\alpha}D - \alpha^m \mathbb{E}\Big[\sum_{t=1}^{m-1} \frac{L}{\alpha^{t+1}}\Big(\frac{1}{2t^2} - \frac{13}{8t}\frac{1}{18\sqrt{n}} - \frac{1}{6b}\Big)\|x_t^s - \widetilde{x}^{s-1}\|^2\Big]$$

$$\leq \alpha^m \mathbb{E}[\Phi(\widetilde{x}^{s-1}) - \Phi^*] + \frac{1-\alpha^m}{1-\alpha}D - \alpha^m \mathbb{E}\Big[\sum_{t=1}^{m-1} \frac{L}{\alpha^{t+1}}\Big(\frac{1}{2t^2} - \frac{1}{8\sqrt{n}t} - \frac{1}{6b}\Big)\|x_t^s - \widetilde{x}^{s-1}\|^2\Big]$$

$$\leq \alpha^m \mathbb{E}[\Phi(\widetilde{x}^{s-1}) - \Phi^*] + \frac{1-\alpha^m}{1-\alpha}D, \qquad (49)$$

where (47) holds since $\|\cdot\|^2$ always be non-negative and $x_0^s = \widetilde{x}^{s-1}$. (48) holds since $\alpha = 1 - \frac{\mu}{18L}$ and the assumption $L/\mu > \sqrt{n}$. (49) holds since it is sufficient to show that $\Gamma_t \geq 0$ for all $1 \leq t < m$, where $\Gamma_t = \frac{1}{2t^2} - \frac{1}{8\sqrt{n}t} - \frac{1}{6b}$. Taking a derivative for $\Gamma_t$, we get $\Gamma'_t = -\frac{1}{t^3} + \frac{1}{8\sqrt{n}t^2} = -\frac{8\sqrt{n}-t}{8\sqrt{n}t^3} < 0$ since $t < m = \sqrt{b} \leq \sqrt{n}$ (note that for other choices of epoch length $m$, the proof is almost the same as that in Appendix A.2). Thus, $\Gamma_t$ decreases in $t$. We only need to show that $\Gamma_m = \Gamma_{\sqrt{b}} \geq 0$, i.e., $\frac{1}{2b} - \frac{1}{8\sqrt{nb}} - \frac{1}{6b} = \frac{1}{3b} - \frac{1}{8\sqrt{nb}} \geq 0$. It is easy to see that this inequality holds since $b \leq n$.

Similarly, let $\widetilde{\alpha} := \alpha^m$ and $\widetilde{\Psi}^s := \frac{\mathbb{E}[\Phi(\widetilde{x}^s) - \Phi^*]}{\widetilde{\alpha}^s}$. Plugging them into (49) and recalling $D := \frac{I\{B<n\}\eta\sigma^2}{B}$, we have

$$\widetilde{\Psi}^s \leq \widetilde{\Psi}^{s-1} - \frac{1}{\widetilde{\alpha}^s}\frac{1-\widetilde{\alpha}}{1-\alpha}\frac{I\{B < n\}\eta\sigma^2}{B}. \qquad (50)$$

Now, we sum up (50) for all epochs $1 \leq s \leq S$ to finish the proof as follows:

$$
\begin{aligned}
\mathbb{E}[\Phi(\widetilde{x}^S) - \Phi^*] &\leq \widetilde{\alpha}^S \mathbb{E}[\Phi(\widetilde{x}^0) - \Phi^*] + \widetilde{\alpha}^S \sum_{s=1}^{S} \frac{1}{\widetilde{\alpha}^s} \frac{1 - \widetilde{\alpha}}{1 - \alpha} \frac{I\{B < n\}\eta\sigma^2}{B} \\
&= \alpha^{Sm} \mathbb{E}[\Phi(\widetilde{x}^0) - \Phi^*] + \frac{1 - \widetilde{\alpha}^S}{1 - \widetilde{\alpha}} \frac{1 - \widetilde{\alpha}}{1 - \alpha} \frac{I\{B < n\}\eta\sigma^2}{B} \\
&\leq \alpha^{Sm} \mathbb{E}[\Phi(\widetilde{x}^0) - \Phi^*] + \frac{1}{1 - \alpha} \frac{I\{B < n\}\eta\sigma^2}{B} \\
&= \left(1 - \frac{\mu}{18L}\right)^{Sm} (\Phi(x_0) - \Phi^*) + \frac{I\{B < n\}18L\eta\sigma^2}{\mu B} \quad\quad (51) \\
&= \left(1 - \frac{\mu}{18L}\right)^{Sm} (\Phi(x_0) - \Phi^*) + \frac{I\{B < n\}3\sigma^2}{\mu B} = 2\epsilon, \quad\quad (52)
\end{aligned}
$$

where (51) holds sine $\alpha = 1 - \frac{\mu}{18L}$, and (52) uses $\eta = \frac{1}{6L}$.

From (52), we obtain the total number of iterations $T = Sm = S\sqrt{b} = O(\frac{1}{\mu} \log \frac{1}{\epsilon})$. The number of PO calls equals to $T = Sm = O(\frac{1}{\mu} \log \frac{1}{\epsilon})$. The number of SFO calls equals to $Sn + Smb = O\left(\frac{n}{\mu\sqrt{b}} \log \frac{1}{\epsilon} + \frac{b}{\mu} \log \frac{1}{\epsilon}\right)$ if $B = n$ (i.e., the second term in (52) is 0 and thus Assumption 1 is not needed), or equals to $SB + Smb = O\left(\frac{B}{\mu\sqrt{b}} \log \frac{1}{\epsilon} + \frac{b}{\mu} \log \frac{1}{\epsilon}\right)$ if $B < n$ (note that $\frac{I\{B<n\}3\sigma^2}{\mu B} \leq \epsilon$ since $B \geq 6\sigma^2/\mu\epsilon$). $\qquad\square$

## B.2 Proof Under Form (8)

**Proof of Theorem 2.** First, we need the following inequality (recall that $\Delta\bar{x}_t^s := \bar{x}_t^s - x_{t-1}^s$):

$$
\begin{aligned}
\Phi(\bar{x}_t^s) &= f(\bar{x}_t^s) + h(\bar{x}_t^s) + h(x_{t-1}^s) - h(x_{t-1}^s) \\
&\leq f(x_{t-1}^s) + \langle \nabla f(x_{t-1}^s), \Delta\bar{x}_t^s \rangle + \frac{L}{2}\|\Delta\bar{x}_t^s\|^2 + h(\bar{x}_t^s) + h(x_{t-1}^s) - h(x_{t-1}^s) \quad (53) \\
&= \Phi(x_{t-1}^s) + \langle \nabla f(x_{t-1}^s), \Delta\bar{x}_t^s \rangle + \frac{L}{2}\|\Delta\bar{x}_t^s\|^2 + h(\bar{x}_t^s) - h(x_{t-1}^s) \\
&\leq \Phi(x_{t-1}^s) + \langle \nabla f(x_{t-1}^s), \Delta\bar{x}_t^s \rangle + \frac{1}{2\eta}\|\Delta\bar{x}_t^s\|^2 + h(\bar{x}_t^s) - h(x_{t-1}^s) \quad (54) \\
&= \Phi(x_{t-1}^s) - \frac{\eta}{2} D_h(x_{t-1}^s, \frac{1}{\eta}) \quad (55) \\
&\leq \Phi(x_{t-1}^s) - \eta\mu(\Phi(x_{t-1}^s) - \Phi^*), \quad (56)
\end{aligned}
$$

where (53) holds since $f$ has $L$-Lipschitz continuous gradient, (54) holds due to $\eta = \frac{1}{6L} < \frac{1}{L}$, (55) follows from the definition of $D_h$ and recall $\bar{x}_t^s := \text{prox}_{\eta h}\left(x_{t-1}^s - \eta\nabla f(x_{t-1}^s)\right)$, and (56) follows from the definition of PL condition with form (8).

Then, adding $\frac{9}{11}$ times (17) and $\frac{2}{11}$ times (56), we have

$$
\begin{aligned}
\Phi(\bar{x}_t^s) &\leq \Phi(x_{t-1}^s) - \frac{9}{11}\left(\frac{1}{\eta} - \frac{L}{2}\right)\|\Delta\bar{x}_t^s\|^2 - \frac{2}{11}\eta\mu(\Phi(x_{t-1}^s) - \Phi^*) \\
&= \Phi(x_{t-1}^s) - \left(\frac{9}{11\eta} - \frac{9L}{22}\right)\|\Delta\bar{x}_t^s\|^2 - \frac{2\eta\mu}{11}(\Phi(x_{t-1}^s) - \Phi^*). \quad (57)
\end{aligned}
$$

We add (57) and (16) to obtain the following inequality (recall that $\Delta x_t^s := x_t^s - x_{t-1}^s$):

$$\Phi(x_t^s) \le \Phi(x_{t-1}^s) + \frac{L}{2}\|\Delta x_t^s\|^2 - \left(\frac{9}{11\eta} - \frac{9L}{22} - \frac{L}{2}\right)\|\Delta \bar{x}_t^s\|^2 - \frac{2\eta\mu}{11}(\Phi(x_{t-1}^s) - \Phi^*)$$
$$- \frac{1}{\eta}\langle \Delta x_t^s, x_t^s - \bar{x}_t^s \rangle + \langle \xi_{t-1}^s, x_t^s - \bar{x}_t^s \rangle$$

$$= \Phi(x_{t-1}^s) + \frac{L}{2}\|\Delta x_t^s\|^2 - \left(\frac{9}{11\eta} - \frac{9L}{22} - \frac{L}{2}\right)\|\Delta \bar{x}_t^s\|^2 - \frac{2\eta\mu}{11}(\Phi(x_{t-1}^s) - \Phi^*)$$
$$- \frac{1}{2\eta}\left(\|\Delta x_t^s\|^2 + \|x_t^s - \bar{x}_t^s\|^2 - \|\Delta \bar{x}_t^s\|^2\right) + \langle \xi_{t-1}^s, x_t^s - \bar{x}_t^s \rangle$$

$$= \Phi(x_{t-1}^s) - \left(\frac{1}{2\eta} - \frac{L}{2}\right)\|\Delta x_t^s\|^2 - \left(\frac{7}{22\eta} - \frac{10L}{11}\right)\|\Delta \bar{x}_t^s\|^2 - \frac{2\eta\mu}{11}(\Phi(x_{t-1}^s) - \Phi^*)$$
$$- \frac{1}{2\eta}\|x_t^s - \bar{x}_t^s\|^2 + \langle \xi_{t-1}^s, x_t^s - \bar{x}_t^s \rangle$$

$$\le \Phi(x_{t-1}^s) - \left(\frac{1}{2\eta} - \frac{L}{2}\right)\|\Delta x_t^s\|^2 - \left(\frac{7}{22\eta} - \frac{10L}{11}\right)\|\Delta \bar{x}_t^s\|^2 - \frac{2\eta\mu}{11}(\Phi(x_{t-1}^s) - \Phi^*)$$
$$- \frac{1}{8\eta}\|\Delta x_t^s\|^2 + \frac{1}{6\eta}\|\Delta \bar{x}_t^s\|^2 + \langle \xi_{t-1}^s, x_t^s - \bar{x}_t^s \rangle \tag{58}$$

$$= \Phi(x_{t-1}^s) - \left(\frac{5}{8\eta} - \frac{L}{2}\right)\|\Delta x_t^s\|^2 - \left(\frac{5}{33\eta} - \frac{10L}{11}\right)\|\Delta \bar{x}_t^s\|^2 - \frac{2\eta\mu}{11}(\Phi(x_{t-1}^s) - \Phi^*)$$
$$+ \langle \xi_{t-1}^s, x_t^s - \bar{x}_t^s \rangle$$

$$\le \Phi(x_{t-1}^s) - \left(\frac{5}{8\eta} - \frac{L}{2}\right)\|\Delta x_t^s\|^2 - \left(\frac{5}{33\eta} - \frac{10L}{11}\right)\|\Delta \bar{x}_t^s\|^2 - \frac{2\eta\mu}{11}(\Phi(x_{t-1}^s) - \Phi^*)$$
$$+ \eta\|\xi_{t-1}^s\|^2. \tag{59}$$

In the same way as (18) and (19), (58) uses Young's inequality (20) (choose $\alpha = 3$) and (59) follows from Lemma 2.

Now, we take expectations for (59) and then plug the variance bound (29) into it to obtain (recall that $D := \frac{I\{B<n\}\eta\sigma^2}{B}$):

$$\mathbb{E}[\Phi(x_t^s)] \le \mathbb{E}\Big[\Phi(x_{t-1}^s) - \left(\frac{5}{8\eta} - \frac{L}{2}\right)\|\Delta x_t^s\|^2 - \left(\frac{5}{33\eta} - \frac{10L}{11}\right)\|\Delta \bar{x}_t^s\|^2 - \frac{2\eta\mu}{11}(\Phi(x_{t-1}^s) - \Phi^*)$$
$$+ \frac{\eta L^2}{b}\|x_{t-1}^s - \widetilde{x}^{s-1}\|^2 + D\Big]$$

$$= \mathbb{E}\Big[\Phi(x_{t-1}^s) - \frac{13L}{4}\|\Delta x_t^s\|^2 - \frac{\mu}{33L}(\Phi(x_{t-1}^s) - \Phi^*) + \frac{L}{6b}\|x_{t-1}^s - \widetilde{x}^{s-1}\|^2 + D\Big] \tag{60}$$

$$\le \mathbb{E}\Big[\Phi(x_{t-1}^s) - \frac{13L}{8t}\|x_t^s - \widetilde{x}^{s-1}\|^2 - \frac{\mu}{33L}(\Phi(x_{t-1}^s) - \Phi^*)$$
$$+ \left(\frac{L}{6b} + \frac{13L}{8t-4}\right)\|x_{t-1}^s - \widetilde{x}^{s-1}\|^2 + D\Big], \tag{61}$$

where (60) uses $\eta = \frac{1}{6L}$, and (61) uses Young's inequality $\|x_t^s - \widetilde{x}^{s-1}\|^2 \le \left(1 + \frac{1}{\alpha}\right)\|x_{t-1}^s - \widetilde{x}^{s-1}\|^2 + (1+\alpha)\|x_t^s - x_{t-1}^s\|^2$ by choosing $\alpha = 2t - 1$.

Now, according to (61), we obtain the following key inequality

$$\mathbb{E}[\Phi(x_t^s) - \Phi^*] \le \mathbb{E}\Big[\left(1 - \frac{\mu}{33L}\right)(\Phi(x_{t-1}^s) - \Phi^*) - \frac{13L}{8t}\|x_t^s - \widetilde{x}^{s-1}\|^2$$
$$+ \left(\frac{L}{6b} + \frac{13L}{8t-4}\right)\|x_{t-1}^s - \widetilde{x}^{s-1}\|^2 + D\Big]. \tag{62}$$

The remaining proof is exactly the same as our proof in Appendix B.1 from (45) to the end. $\qquad\square$