[Reviews · NeurIPS 2018]

Reviewer 1



This paper focuses on the optimization problem min f(x) + h(x), where f is of a finite sum structure (with n functions in the sum), with nonconvex but smooth components, and h is a convex but possibly nonsmooth function. So, this is a nonconvex finite sum problem with a convex regularizer. Function h is treated using a prox step. The authors propose a small modification to ProxSVRG (called ProxSVRG+), and prove that this small modification has surprisingly interesting consequences. The modification consists in replacing the full gradient computation in the outer loop of ProxSVRG by an approximation thereof through subsampling/minibatch (batch size B). In the inner loop, minibatch steps are run (with minibatch size b). Main theoretical results: • The SFO complexity (i.e., number of calls of stochastic first order oracle) of ProxSVRG+ is better than the complexity of ProxSVRG, ProxSGD and ProxGD in all minibatch regimes (any b). The results are nicely summarized in Tables 1, 2 and Figure 1 on page 3. The complexity is O( n/(eps*sqrt{b}) + b/eps ) under no additional assumptions. • Further improvement is obtained when one assumes bounded variance of the stochastic gradients, with sigma = O(1). The improvement consist in replacing the “n” in the complexity by the minimum of n and 1/eps. • The SFO complexity is better than that of SCSG in the regimes when b is not too small and not too large. • ProxSVRG results hold for b <= n^{2/3}, and ProxSGD results hold for b >= 1/eps. However, ProxSVRG+ result holds for all values of b. • The PO complexity (number of proximal steps) is O(1/\eps), which matches the best PO complexity of ProxGD and ProxSGD, and is better than the PO complexity of Natasha1.5 and SCSG. Additional results of similar nature (but with logarithmic dependence on epsilon, as expected) are obtained under the PL (Polyak-Lojasiewicz) condition. A new variant of PL condition is proposed and used in the analysis. Unlike ProxSVRG, no restart is needed for ProxSVRG+. Computational results are provided to support the theory. Strengths: The paper is well-written (there is only a small number of grammatical issues) and well-motivated. The fact that only a minor modification of ProxSVRG is needed to obtain all the improvements should not be seen negatively. The new analysis with the modification is simpler, more elegant, and tighter. The computational experiments clearly show that ProSVRG+ can obtain similar performance as ProxSVRG with much smaller minibatch sizes b (order of magnitude smaller). This is both interesting and important. This is a very nice paper that should be accepted. Weaknesses and Questions: 1) Can the results be improved so that ProxSVRG+ becomes better than SCSG for all minibatch sizes b? 2) How does the PL condition you use compare with the PL conditions proposed in “Global Convergence of Arbitrary-Block Gradient Methods for Generalized Polyak-Łojasiewicz Functions”, arXiv:1709.03014 ? 3) Line 114: Is the assumption really necessary? Why? Or just sufficient? 4) I think the paper would benefit if some more experiments were included in the supplementary, on some other problems and with other datasets. Otherwise the robustness/generalization of the observations drawn from the included experiments is unclear. Small issues: 1) Lines 15-16: The sentence starting with “Besides” is not grammatically correct. 2) Line 68: What is a “super constant”? 3) Line 75: “matches” -> “match” 4) Page 3: “orcale” – “oracle” (twice) 5) Caption of Table 1: “are defined” -> “are given” 6) Eq (3): use full stop 7) Line 122: The sentence is not grammatically correct. 8) Line 200: “restated” -> “restarted” 9) Paper [21]: accents missing for one author’s name ***** I read the rebuttal and the other reviews, and am keeping my score.

Reviewer 2



The proposed ProxSVRG+ generalizes and extends previous SCSG and ProxSVRG. Using a new convergence analysis compared to ProxSVRG, the paper shows that ProxSVRG+ is faster than ProxGD even for small mini-batch size. Besides, a moderate mini-batch size can be needed for ProxSVRG+ to achieve the best convergence result. This provides a solution to an open problem from ProxSVRG paper. At last, the paper proves linear convergence rate of nonconvex function satisfying PL condition without restart, unlike ProxSVRG. Overall, the paper is well-written and easy to follow. Figure 1 is informative to show the relation among ProxSVRG+ and other related algorithms. The experiments shows clear the advantage of ProxSVRG+ and its preference of using small mini-batch size. Regarding the paper, I have the following concerns 1. The paper argues theoretically that the best mini-batch size required for ProxSVRG+ is 1/{\epsillon^{2/3}}, where \epsillon is the convergence criterion. Considering the claims in the comparison with ProxSVRG, \epsilon should be greater than 1/n for a smaller mini-batch size, would it be a problem to claim a good convergence for ProxSVRG+? 2. As in the ProxSVRG+ Algorithm, a subsample gradient control step is used (step 3-5) with the batch-size B. However, in the experiment, B is not mentioned. could you specify the B you use in your experiments and how the performance of ProxSVRG+ is changed regarding different B. (please refer to SCSG paper.) 3. At line 225, it says ProxSVRG+ always performs better than others, is it overclaimed, due to you are using fixed step size (for example ProxSGD is suggested to have diminished step-size rather than a constant one. please refer to ProxSVRG paper) for all algorithms and the mini-batch size are limited to 256, which is later on claimed as the best for ProxSVRG+. minor: 1. In Algorithm 1, T is not defined and used. And S should not be so-called number of epochs (1 epoch = n SFOs)? or please define it.

Reviewer 3



## Summary of paper The authors provide a novel analysis of a minor variant of the variance reduced algorithm (SVRG) for the composite setting--proxSVRG+. The main contribution is a new simpler analysis which recovers most previous convergence rates, and even improve upon them in certain regimes of the data-size and accuracy required. Further, they analyze the case when arbitrary batch sizes are used at each iteration of prox-SVRG instead of the full batch, and theoretically show that smaller batches can be optimal when the variance of the stochastic gradient is small and we do not require a high accuracy solution. ## Significane of results This work has results which may perhaps be significant for practitioners--i) the smaller mini-batch requirement, and ii) adaptivity to strong convexity. Theoretically, though the authors unify a number of previous results, they do not recover the rates by the SCSG when $h=0$ indicating that the prox-SVRG algorithm requires further study. ## The positives 1. The paper is well written and is easy to read--the authors make an effort explaining previous result and how their result compares to related work (e.g. pp 3) 2. The analysis indeed seems simpler (or is at-least more *standard*) than previous works. 3. The extension to the case when a smaller batch (B) is used seems very useful in practice. The authors confirm this experimentally where they show that the optimal minibatch size of their algorithm is indeed small. 4. Further the authors show that the algorithm is adaptive to strong convexity (the PL-condition), which is important for faster convergence in practice. ## Negatives 1. The metric used to define accuracy (Def 1) is not algorithm independent! In particular it depends on the step-size the particular algorithm uses. This means that technically the results comparing different algorithms are meaningless because the error metric used is different. 2. The experimental section lacks a lot of very important details--i) what iterate is being compared in the different algorithms? For the prox-SVRG+, is it the randomly chosen one for which the theoretical results are proved? How is the $B$ parameter set? Since the 'default' parameters without tuning are used for the other algorithms, it would be an unfair comparison if the $B$ parameter was tuned for the current algorithm. Other minor points: 3. In Fig 1. Natasha is not plotted, so harder to compare. 4. Lemma 2 in the remark (143, pp 5) is mentioned but not stated. In general comments such as ones mentioned below in the Remark on pp. 5 seem vacuous and could potentially be improved to provide meaningful insights into the proof technique. "This is made possible by tightening the inequalities in several places, e.g., using Young’s inequality on different terms and applying Lemma 2 in a nontrivial way" ## What would increase my score 1. Clarify how error as defined in Def. 1 is a meaningful metric to use for comparison. I understand past works may have used it without a second thought, but I find this unsatisfying. 2. Demonstrate that the comparisons performed in the experiments were indeed fair (wrt B) ## Edits after rebuttal 1. The authors do a good job of addressing the meaningfulness of the metric. It would be nice if this discussion in the rebuttal is added to the paper too. 2. The fact that B is set to n/5 and need not be tuned is enough to convince me that the experiments are valid. I would however appreciate more details about the experimental setup in the final version to improve the replicability. 3. The rates of SCSG are indeed recovered by ProxSVRG+--thanks to the authors for pointing out my oversight.